# The Pareto-optimal Trade-off between Regret and Statistical Inference in Linear Stochastic Bandits under Safety Constraints

**Yuming Shao** [1 2]  **Zhixuan Fang** [1 3]

## Abstract

Linear bandits traditionally prioritize regret minimization, often overlooking statistical inference of the underlying parameter as a critical objective. In high-stakes settings such as healthcare, precise parameter estimation is indispensable, as it provides fundamental insights into system mechanisms and ensures robust decision-making under covariate shift. We investigate the tripartite balance between regret, inference, and safety, deriving a fundamental minimax lower bound that characterizes the Pareto-optimal frontier of these competing goals. We then propose SERMiSC, a novel algorithm that achieves the optimal trade-off by matching this lower bound while maintaining a near-constant $\tilde{O}(1)$ safety risk. Empirical results demonstrate that SERMiSC effectively navigates the Pareto frontier and outperforms various baselines, thereby validating our theoretical analysis.

## 1. Introduction

The linear stochastic bandit problem has emerged as a pivotal framework for online decision-making under uncertainty, with applications ranging from mobile health (Huch et al., 2024) to algorithmic trading (Ji et al., 2022). In this model, a decision-maker chooses an action represented by a $d$-dimensional feature vector at each time step, receiving a reward that is a linear function of the chosen feature and an unknown parameter $\theta^*$. Traditionally, the primary objective in this literature has been regret minimization—the pursuit of maximizing cumulative rewards by balancing the exploration of new actions with the exploitation of currently estimated high-performing ones (Abbasi-Yadkori et al., 2011).

In online learning scenarios such as clinical trials, minimiz-

ing regret undeniably constitutes an operational imperative to ensure that the welfare of the current patient cohort is not compromised. However, focusing exclusively on cumulative performance often overlooks the critical role of statistical inference. Estimating the underlying $\theta^*$ is essential because it directly characterizes the Average Treatment Effect (ATE), a pivotal metric in clinical research. More importantly, uncovering the intrinsic mechanisms governed by $\theta^*$ is indispensable for generalizing findings to broader populations and ensuring reliability against future shifts in patient demographics (Villar et al., 2015). Consequently, the challenge lies in conducting rigorous inference without disrupting the clinical efficacy of the ongoing process.

Beyond these operational goals, safety remains the bedrock of real-world deployment; any performance or inferential gain is rendered moot if safety is compromised. Thus, this dual pursuit must be fundamentally anchored in the prerequisite of safety. Despite its paramount importance, existing literature has rarely investigated the tripartite balance between cumulative performance (regret), parameter estimation (inference), and the requirement of safety. While recent advancements have begun to incorporate statistical inference into the Multi-Armed Bandit (MAB) setting (Simchi-Levi & Wang, 2025), research within the linear stochastic bandit framework remains limited, with few exceptions exploring linear trends (Simchi-Levi et al., 2023). This leaves a significant gap in our understanding of how to navigate the Pareto-optimal frontier (Simchi-Levi & Wang, 2025)—the boundary where one objective cannot be improved without degrading the others—under stringent safety constraints.

**Contributions.** In this paper, we bridge these gaps by investigating the interplay between regret, statistical inference, and safety within a unified framework. Our primary contributions are summarized as follows:

1. *Information-Theoretic Lower Bounds.* We derive a fundamental minimax lower bound on the statistical estimation error under prescribed requirements on regret and safety, providing a theoretical characterization of the Pareto-optimal trade-off between these objectives.

2. *Pareto-optimal Algorithm Design.* We propose SERMiSC, a novel two-phase algorithm. We rigorously

[1]Institute for Interdisciplinary Information Sciences, Tsinghua University, Beijing, China [2]Xiongan AI Institute, Hebei, China [3]Shanghai Qi Zhi Institute, Shanghai, China. Correspondence to: Zhixuan Fang <zfang@mail.tsinghua.edu.cn>.

*Proceedings of the 43rd International Conference on Machine Learning*, Seoul, South Korea. PMLR 306, 2026. Copyright 2026 by the author(s).

prove that while SERMiSC incurs a near-constant safety risk of order $\tilde{O}(1)$, it achieves the optimal Pareto trade-off between regret and statistical inference, precisely matching our theoretical lower bound.

3. *Empirical Validation.* Simulation results show that SERMiSC outperforms baselines by effectively balancing the tripartite objectives, thereby validating our theoretical findings.

## 1.1. Related Works

**Trade-off between Regret and Inference.** Standard bandit literature typically focuses on either regret minimization (Abbasi-Yadkori et al., 2011) or pure exploration (Soare et al., 2014). Recently, Simchi-Levi & Wang (2025) pioneered the study of the fundamental conflict between these two objectives: cumulative regret minimization and statistical estimation precision. Although their primary estimation target is the ATE within a potential outcomes framework (Rubin, 1974), this trade-off has since gained significant traction. It has been extended to various settings, including non-stationary environments where expected rewards evolve over time (Simchi-Levi et al., 2023), network interference where treatments affect the outcomes of interconnected individuals (Zhang & Wang, 2025), and high-dimensional regimes (Duan et al., 2024).

**Safety-aware Linear Bandits.** Ensuring operational safety has emerged as an indispensable requirement in linear bandit applications, leading to several distinct formulations of safety constraints. (i) *High-probability constraints* (Amani et al., 2019; 2020) require every action to remain within a safe region with high probability, typically employing an expansion strategy starting from a known seed set. (ii) *Conservativeness-style constraints* focus on performance-based safety, either by ensuring that the cumulative reward stays above a baseline percentage (Kazerouni et al., 2017) or by maintaining stagewise expected rewards above a threshold (Khezeli & Bitar, 2020). (iii) *Expectation-based constraints* (Liu et al., 2021; Ma et al., 2024) serve as a relaxation of strict high-probability requirements by focusing on average performance. While the existing safety-aware literature largely neglects the objective of statistical inference, our work addresses this gap by investigating the tripartite balance between regret, inference, and safety.

*Due to space constraints, a more comprehensive discussion of the related literature is provided in Appendix A.*

## 2. Preliminaries

### 2.1. Notation

For any integer $N \in \mathbb{Z}_+$, let $[N] = \{1, \ldots, N\}$. We use boldface letters (e.g., $\boldsymbol{\mu}, \boldsymbol{c}$) to represent vectors, and $\Delta_K$ denotes the $K$-dimensional simplex. For vectors $\boldsymbol{u}, \boldsymbol{v} \in \mathbb{R}^d$, $\langle \boldsymbol{u}, \boldsymbol{v} \rangle$ denotes the standard inner product, and $\boldsymbol{u} \odot \boldsymbol{v}$ represents the element-wise (Hadamard) product. The Kullback-Leibler divergence between two distributions $P$ and $Q$ is denoted by $\text{KL}(P\|Q)$.

### 2.2. Problem Formulation

In this section, we introduce our stochastic linear bandit setup. Within a time horizon $T$, our algorithm needs to make a choice from a fixed set of $K$ arms at the beginning of each time slot. Each arm $a \in [K]$ has a $d$-dimensional feature vector $x_a$, which is associated with its reward distribution. The expected reward of arm $a$ is the inner product between its feature vector $x_a$ and an unknown underlying true parameter vector $\theta^*$, where $\theta^*$ lies in a reasonably bounded subset $\mathcal{B}_\theta \subset \mathbb{R}^d$. Formally, we consider the reward at time slot $t \in [T]$ as being generated by

$$y_t = \langle x_{a_t}, \theta^* \rangle + \eta_t,$$

where $a_t \in [K]$ is the algorithm's arm choice at time $t$ and $\{\eta_t\}_{t \in [T]}$ is a sequence of conditionally $\sigma$-subGaussian stochastic noise, i.e.,

$$\mathbb{E}\Big[ \exp(\psi \eta_t) \Big| \mathcal{F}_{t-1} \Big] \leq \exp\Big( \frac{\psi^2 \sigma^2}{2} \Big), \quad \forall \psi \in \mathbb{R}, t \geq 1,$$

where $\mathcal{F}_{t-1}$ is the filtration $\sigma(a_1, y_1, ..., a_{t-1}, y_{t-1}, a_t)$. We have that $\mathbb{E}[y_t|\mathcal{F}_{t-1}] = \langle x_{a_t}, \theta^* \rangle$. Define $\mu_a = \langle x_a, \theta^* \rangle$ for any $a \in [K]$.

In realistic scenarios, selecting different arms not only affects the reward received but also introduces varied risks, such as those related to safety and budget. Consequently, we incorporate a general cost associated with each arm and establish constraints that the algorithm must strive to satisfy. Specifically, we assume that pulling arm $a \in [K]$ at time $t$ incurs a stochastic cost $Z_t(a)$, which is an independent random variable with a (possibly unknown) mean $c_a \in \mathbb{R}$, and consider the following cumulative safety constraint, defined for each $\tau \in [T]$ (with $T$ denoting the total horizon):

$$\mathcal{C}(\tau) := \mathbb{E}\left[ \sum_{t=1}^{\tau} \sum_{a \in [K]} Z_t(a) \pi_t(a) \right] \leq 0,$$

following Liu et al. (2021) and Ma et al. (2024). $\boldsymbol{\pi}_t \in \Delta^K$ denotes the arm selection policy at time $t$. We can also loosen this constraint by replacing zero with a violation tolerance parameter $\delta_\tau > 0$.

In contrast to the standard linear stochastic bandit, our objective involves a dual consideration: maximizing the cumulative reward and ensuring that the algorithm can provide an accurate estimate of the true parameter vector $\theta^*$. To define the performance metrics of an algorithm, we generally need

a baseline policy against which we measure performance. Our algorithm's efficacy is defined by how far its decisions deviate from this baseline.

Let $a^* \in \arg\max_{a \in [K]} \langle x_a, \theta^* \rangle$ denote the optimal arm. In the standard (unconstrained) scenario, the fixed policy $\boldsymbol{\pi}$, where $\pi_t(a^*) = 1$ for all $t \in [T]$, serves as the baseline, reflecting our goal of pulling the optimal arm as often as possible. Nonetheless, when cost constraints are introduced, a policy that only maximizes reward might result in a positive expected cost, thereby violating the constraints. The fixed policy $\boldsymbol{\pi}$ is obviously unacceptable in our scenario. Instead, we adopt as our baseline

$$\boldsymbol{\pi}^* \in \arg\max_{\boldsymbol{\pi} \in \Delta_K} \big\{ \langle \boldsymbol{\pi}, \boldsymbol{\mu} \rangle \mid \langle \boldsymbol{\pi}, \boldsymbol{c} \rangle \leq 0 \big\},$$

with $\boldsymbol{\mu} = (\mu_a)_{a \in [K]}$ and $\boldsymbol{c} = (c_a)_{a \in [K]}$ collecting the expected rewards and costs introduced above. More generally, we define the optimal value of the safety-constrained reward maximization problem as

$$\mathcal{P}(\gamma) := \max_{\boldsymbol{\pi} \in \Delta_K} \big\{ \langle \boldsymbol{\pi}, \boldsymbol{\mu} \rangle \mid \langle \boldsymbol{\pi}, \boldsymbol{c} \rangle + \gamma \leq 0 \big\}.$$

Intuitively, $\boldsymbol{\pi}^*$ represents the optimal static policy that maximizes the expected reward while satisfying the safety constraint.

Thus, we evaluate the performance of our algorithms in terms of the following two metrics:

- **Cumulative Regret.** The expected cumulative difference between the optimal single-round reward and the actual reward

$$\mathcal{R}(\tau) = \mathbb{E}\left[ \sum_{t=1}^{\tau} \sum_{a \in [K]} \pi_a^* \cdot \langle x_a, \theta^* \rangle \right.$$
$$\left. - \sum_{t=1}^{\tau} \langle x_{a_t}, \theta^* \rangle \right].$$

- **Statistical Estimation Error.** The norm of the difference between the algorithm's statistical estimate and the ground truth

$$err(\tau) = \mathbb{E}\big[ ||\hat{\theta}_\tau - \theta^*||_2 \big],$$

where $\hat{\theta}_t$ is the estimator for the true parameter $\theta^*$ constructed by the algorithm at time $t$ using the historical observations.

## 3. Pareto Optimal Trade-off

As established in the preceding section, the problem we address is fundamentally a constrained multi-objective optimization task. In such scenarios, the existence of a single

"utopian" solution that simultaneously optimizes all conflicting objectives is a rarity. Consequently, we adopt the framework of **Pareto optimality**. Rather than seeking a singular optimum, Pareto optimality characterizes the boundary of efficiency: the Pareto Frontier. On this frontier, no individual objective (e.g., estimation accuracy) can be improved without a commensurate degradation in at least one other objective (e.g., cumulative reward or safety). This concept allows us to rigorously quantify the theoretical cost of precision: specifically, how much additional regret or safety violation must be incurred to achieve a marginal gain in statistical estimation accuracy.

We begin this section by introducing the intuition behind the inherent tension between three distinct objectives: statistical estimation, regret minimization, and safety constraint satisfaction. Subsequently, we derive a theoretical Pareto-optimal trade-off specific to the safety-constrained linear bandit setting under consideration.

To grasp the inherent tension between statistical estimation and regret minimization, we first examine the limitations of the reward-maximizing policy. Let $\mathcal{K}^* = \{a \in [K] : \pi_a^* > 0\}$ denote the support of the optimal safe policy $\boldsymbol{\pi}^*$. While $\boldsymbol{\pi}^*$ is ideal for minimizing regret and satisfying safety constraints, the set of actions $\mathcal{K}^*$ may span only a low-dimensional subspace $\text{span}(\mathcal{K}^*)$ of the full parameter space $\mathbb{R}^d$. Consequently, its associated design matrix $\Sigma(\boldsymbol{\pi}^*) = \sum_{a \in [K]} \pi_a^* x_a x_a^\top$ can be rank deficient. In other words, $\boldsymbol{\pi}^*$ provides insufficient geometric information to recover the full parameter $\theta^*$.

To mitigate this issue, a reasonable strategy must shift the probability mass of $\boldsymbol{\pi}^*$ towards an exploratory direction, denoted by an auxiliary policy $\boldsymbol{\pi}'$ with $\lambda_{\min}\big(\Sigma(\boldsymbol{\pi}')\big) > 0$. By augmenting $\boldsymbol{\pi}^*$ with $\boldsymbol{\pi}'$, the resulting policy $\boldsymbol{\pi}$ can reach the previously hidden dimensions of the parameter space, thereby resolving the rank deficiency of the design matrix and enhancing its estimation power. Formally, we consider

$$\boldsymbol{\pi} = (1 - \kappa)\boldsymbol{\pi}^* + \kappa \boldsymbol{\pi}', \tag{1}$$

where $\kappa \in (0, 1)$ represents the offset magnitude (or exploration budget). This decomposition allows us to analyze the "price of information" through the lens of $\kappa$: while a larger $\kappa$ leads to a theoretical increase in cumulative regret and safety violations, such that

$$\mathcal{R}(T) \approx T \sum_{a \in [K]} \mu_a(\pi_a^* - \pi_a) = \kappa \cdot T \sum_{a \in [K]} \mu_a(\pi_a^* - \pi_a'),$$

$$Vio(T) \gtrsim T \sum_{a \in [K]} c_a \pi_a = \kappa \cdot T \sum_{a \in [K]} c_a \pi_a',$$

it simultaneously facilitates a lower limit on the achievable

statistical estimation error:

$$err(T) \gtrsim \frac{1}{\sqrt{T\lambda_{\min}(\Sigma(\boldsymbol{\pi}))}} = \frac{1}{\sqrt{\kappa \cdot T\lambda_{\min}(\Sigma(\boldsymbol{\pi}'))}}.$$

The preceding analysis reveals that the parameter $\kappa$ effectively traces the Pareto frontier of the safety-constrained linear bandit problem. Having established this intuition, we now proceed to formalize this relationship. In the remainder of this section, we state and prove our main theoretical result. This result establishes a fundamental lower bound on the estimation error relative to the allowed cumulative regret and safety violation, thereby providing a rigorous characterization of the Pareto optimal trade-off.

**Theorem 3.1** (Pareto Minimax Lower Bound on the Statistical Estimation Error). *Consider the safety-constrained linear bandit setup with fixed action set $\mathcal{X} = \{x_a\}_{a\in[K]}$ and known subGaussian parameter $\sigma$. An environment $\nu$ is fully characterized by the pair $(\theta^*, \boldsymbol{c})$ of the parameter vector and the expected cost vector; we denote the environment class by $\mathcal{V} := \mathcal{B}_\theta \times \mathcal{B}_c$, where $\mathcal{B}_c \subset \mathbb{R}^K$ is a bounded set containing the cost vectors of interest, and write $\theta^*_\nu := \theta^*$ for any $\nu = (\theta^*, \boldsymbol{c}) \in \mathcal{V}$. An* estimator *at time $\tau \in [T]$ is any measurable mapping $\hat{\theta}_\tau$ from the observation history $(a_1, y_1, \ldots, a_\tau, y_\tau)$ to $\mathbb{R}^d$; let $\widehat{\Theta}$ denote the class of all such estimators.*

*For $\beta \in (0, 1)$ and a tolerance parameter $\delta_T \geq 0$, define the admissible policy class, which captures all algorithms achieving sublinear regret of order $T^\beta$ while keeping the safety violation within budget $\delta_T$:*

$$\Pi(\beta, \delta_T) := \left\{ \boldsymbol{\pi} \quad s.t. \begin{cases} \mathcal{R}(T) = O(T^\beta) \\ \mathcal{C}(T) = O(\delta_T) \end{cases} \right\}.$$

*Then there exist constants $\sigma', C > 0$ (independent of $T$) such that*

$$\inf_{\substack{\boldsymbol{\pi} \in \Pi(\beta, \delta_T) \\ \hat{\theta} \in \widehat{\Theta}}} \sup_{\nu \in \mathcal{V}} \mathbb{E}^{\boldsymbol{\pi}}_\nu \left[ \|\hat{\theta} - \theta^*_\nu\|_2 \right] \geq \frac{5\sigma'}{2\sqrt{\delta_T + 10CT^\beta}}.$$

(2)

Here, we give a sketch of the proof of Theorem 3.1. A complete proof can be found in Appendix B.

*Proof Sketch of Theorem 3.1.* The proof consists of the following three steps.

**Step 1: Information-Theoretic Reduction to KL Divergence**

In the minimax formulation, the supremum over the set of all possible environments can be relaxed by considering a subset of two specifically constructed environments, $\nu_1$ and $\nu_2$. These environments are designed to be statistically difficult to distinguish for any admissible policy. The parameter vectors for these environments are defined as:

$$\theta^*_1 = \begin{bmatrix} 1 \\ 0 \end{bmatrix}, \quad \theta^*_2 = \begin{bmatrix} 1 \\ 2\Delta \end{bmatrix},$$

where $\Delta > 0$ is a small perturbation constant. Furthermore, we define an action set consisting of the following three arms:

$$x_1 = \begin{bmatrix} 0.2 \\ 0.0 \end{bmatrix}, \qquad x_2 = \begin{bmatrix} 0.4 \\ 0.0 \end{bmatrix}, \qquad x_3 = \begin{bmatrix} 0.1 \\ 0.1 \end{bmatrix},$$
$$c_1 = -1, \qquad c_2 = 1, \qquad c_3 = 2.$$

This construction ensures that in environment $\nu_1$,

1. The baseline policy $\boldsymbol{\pi}^*$ is $[0.5, 0.5, 0.0]^\top$, and its associated design matrix $\Sigma(\boldsymbol{\pi}^*)$ is rank deficient, as it only spans the first dimension.

2. There exists an auxiliary exploratory policy $\boldsymbol{\pi}'$ (e.g., $[1/3, 1/3, 1/3]^\top$) that yields an invertible design matrix by incorporating arm 3, which spans the second dimension.

As established earlier in this section, this specific problem instance necessitates a trade-off between statistical estimation error and regret/safety penalties. Then, we show that it suffices to upper bound the KL divergence between the probability measures associated with both environments, utilizing techniques such as the Neyman-Pearson lemma (Lehmann & Romano, 2005) and Pinsker's inequality (Cover, 1999):

$$\inf_{\substack{\boldsymbol{\pi} \in \Pi(\beta, \delta_T) \\ \hat{\theta} \in \widehat{\Theta}}} \sup_{\nu \in \mathcal{V}} \mathbb{P}^{\boldsymbol{\pi}}_\nu \left( \|\hat{\theta} - \theta^*_\nu\|_2 \geq \Delta \right)$$

$$\geq \inf_{\substack{\boldsymbol{\pi} \in \Pi(\beta, \delta_T) \\ \hat{\theta} \in \widehat{\Theta}}} \max_{i \in \{1,2\}} \mathbb{P}^{\boldsymbol{\pi}}_{\nu_i} \left( \|\hat{\theta} - \theta^*_i\|_2 \geq \Delta \right)$$

$$\geq \frac{1}{2} \left[ 1 - \sqrt{\frac{1}{2} \sup_{\boldsymbol{\pi} \in \Pi(\beta, \delta_T)} \mathrm{KL}(\mathbb{P}^{\boldsymbol{\pi}}_{\nu_1} \| \mathbb{P}^{\boldsymbol{\pi}}_{\nu_2})} \right],$$

where $\mathbb{P}^{\boldsymbol{\pi}}_\nu \left( \|\hat{\theta} - \theta^*_\nu\|_2 \geq \Delta \right)$ is the error probability closely related to the statistical estimation error.

**Step 2: Linear Programming Formulation of the KL Bound**

Evaluating $\sup_{\boldsymbol{\pi} \in \Pi(\beta, \delta_T)} \mathrm{KL}(\mathbb{P}^{\boldsymbol{\pi}}_{\nu_1} \| \mathbb{P}^{\boldsymbol{\pi}}_{\nu_2})$ requires a maximization over all admissible policies in $\Pi(\beta, \delta_T)$. To simplify this procedure, we follow (Lattimore & Szepesvári, 2020) and decompose the overall KL divergence such that

$$\mathrm{KL}(\mathbb{P}^{\boldsymbol{\pi}}_{\nu_1} \| \mathbb{P}^{\boldsymbol{\pi}}_{\nu_2})$$
$$= \sum_{t=1}^{T} \sum_{a \in [K]} \mathrm{KL}_a(P^{\boldsymbol{\pi}}_{\nu_1,t} \| P^{\boldsymbol{\pi}}_{\nu_2,t}) P^{\boldsymbol{\pi}}_{\nu_1,t}(a_t = a),$$

where $P_{\nu,t}^{\boldsymbol{\pi}}$ is the probability measure with respect to time slot $t$. By expressing the constraints $\mathcal{R}(T) = O(T^\beta)$ and $\mathcal{C}(T) = O(\delta_T)$ in terms of the sequence of measures $\{P_{\nu_1,t}^{\boldsymbol{\pi}}\}_{t \in [T]}$ and exploiting the inherent temporal symmetry of the problem, we can recast the search for the KL divergence upper bound as the following linear program.

$$\max_{\tilde{\boldsymbol{\pi}}} \quad \sum_{a \in [K]} \mathrm{KL}_a(P_{\nu_1,T}^{\boldsymbol{\pi}} \| P_{\nu_2,T}^{\boldsymbol{\pi}}) \tilde{\pi}_a T,$$

$$s.t. \quad \sum_{a \in [K]} c_a \tilde{\pi}_a T \leq \delta_T,$$

$$\sum_{a \in [K]} \mu_a(\pi_a^* - \tilde{\pi}_a) T \leq CT^\beta,$$

$$\tilde{\boldsymbol{\pi}} \in \Delta_K.$$

### Step 3: Establishing the Bound via Dual Variable Design

Starting from the linear program derived in the previous step, we apply standard Lagrangian duality by introducing dual variables (e.g., $\boldsymbol{\lambda} \succeq 0$ to account for the safety constraints and the regret). We can transform the primal maximization of the KL divergence into the following dual form

$$\min_{\boldsymbol{\lambda} \succeq 0} \quad \lambda_1 \delta_T + \lambda_2 CT^\beta + T \cdot \max_{a' \in [K]} \iota_{a'},$$

where

$$\iota_{a'} := \mathrm{KL}_{a'}(P_{\nu_1,T}^{\boldsymbol{\pi}} \| P_{\nu_2,T}^{\boldsymbol{\pi}})$$
$$- \lambda_1 c_{a'} - \lambda_2 \Big( \sum_{a \in [K]} \mu_a \pi_a^* - \mu_{a'} \Big).$$

We proceed with the derivation by appropriately assigning values to the dual variables $\lambda_1$ and $\lambda_2$. To ensure that the resulting upper bound is non-trivial, the coefficient of $T$ in the final term must be non-positive. Under our problem instance construction, the two environments $\nu_1$ and $\nu_2$ are indistinguishable with respect to the first two arms, i.e., $\mathrm{KL}_a(P_{\nu_1,T}^{\boldsymbol{\pi}} \| P_{\nu_2,T}^{\boldsymbol{\pi}}) = 0$ for $a \in \{1, 2\}$. Furthermore, given that $c_1 c_2 < 0$ and $\frac{c_1}{c_2} = \frac{\sum_{a \in [K]} \mu_a \pi_a^* - \mu_1}{\sum_{a \in [K]} \mu_a \pi_a^* - \mu_2}$, the only way to satisfy $\max_{a \in \{1,2\}} \iota_a \leq 0$ is to set the ratio $\frac{\lambda_1}{\lambda_2} = \frac{\mu_2 - \mu_1}{c_2 - c_1}$. By imposing the additional requirement $\iota_3 = 0$, we obtain a unique feasible assignment for $\boldsymbol{\lambda}$. The existence of such a feasible $\boldsymbol{\lambda}$ is guaranteed by the fact that $c_3 > 0$ and $\sum_{a \in [K]} \mu_a \pi_a^* - \mu_3 > 0$. This assignment for $\boldsymbol{\lambda}$ yields the upper bound

$$\sup_{\boldsymbol{\pi} \in \Pi(\beta, \delta_T)} \mathrm{KL}(\mathbb{P}_{\nu_1}^{\boldsymbol{\pi}} \| \mathbb{P}_{\nu_2}^{\boldsymbol{\pi}}) = O\Big(\Delta^2 (\delta_T + CT^\beta)\Big).$$

With $\Delta = \Theta\big(\frac{1}{\sqrt{\delta_T + CT^\beta}}\big)$, the proof of (2) follows from direct calculation. $\square$

---

**Algorithm 1** SERMiSC - **Phase I:** Statistical Estimation

1: **Input:** $\boldsymbol{\pi}'$, $q_t$, $\zeta_t$
2: $Y, N, \hat{C} \leftarrow [0, ..., 0]^\top \in \mathbb{R}^K$
3: **for** $t = 1, ..., T_1$ **do**
4:     sample and pull the arm $a_t \sim \boldsymbol{\pi}_t$ where

$$\pi_t(i) = q_t \cdot \pi'(i) + (1 - q_t) \cdot \frac{e^{-\zeta_t \hat{C}(i)}}{\sum_{j \in [K]} e^{-\zeta_t \hat{C}(j)}}$$

5:     observe the reward $y_t$
6:     $Y_{a_t} \leftarrow Y_{a_t} + y_t$
7:     $N_{a_t} \leftarrow N_{a_t} + 1$
8:     $\hat{C}(a_t) \leftarrow \hat{C}(a_t) + \frac{Z_t}{\pi_t(a_t)}$
9: **end for**
10: $\mathbb{X}_{\text{stat}} \leftarrow \left[ \sqrt{N_1} x_1, ..., \sqrt{N_K} x_K \right]^\top$
11: $\tilde{Y} \leftarrow Y \odot 1/\sqrt{\max\{N, 1\}}$
12: $\hat{\theta}_{\text{stat}} \leftarrow (\mathbb{X}_{\text{stat}}^\top \mathbb{X}_{\text{stat}})^{-1} \mathbb{X}_{\text{stat}}^\top \tilde{Y}$
13: **Output:** $\hat{\theta}_{\text{stat}}$

---

## 4. Algorithm Design

We propose the Statistical Estimation and Regret Minimization under Safety Constraint (SERMiSC) algorithm, a novel two-phase approach for the joint optimization of our distinct objectives.

The design of this algorithm is motivated by the decomposition presented in (1). Specifically, the algorithm seeks to strategically allocate the sampling budget between the baseline policy $\pi^*$, relative to which regret is defined, and an exploratory policy $\pi'$. The latter ensures sufficient exposure of each arm to maintain a controllable statistical estimation error for $\theta^*$. Consequently, the SERMiSC algorithm is structured into two phases, each striving to approximate policies $\pi'$ and $\pi^*$, respectively, while adhering to safety constraints.

### 4.1. Phase I: Safe Statistical Estimation

Let $T_1$ denote the temporal length of Phase I. The implementation of the SERMiSC algorithm during Phase I is summarized in Algorithm 1. The primary objective of this phase is to obtain a statistically efficient estimate of the parameter vector $\theta^*$ subject to safety constraints. To minimize the estimation error, it is essential to employ an exploratory policy $\pi'$ that ensures sufficient and balanced coverage of the action space.

However, a fundamental challenge arises: since the true cost vector $\boldsymbol{c}$ is unknown a priori, we cannot, at the outset, guarantee that the exploratory policy $\pi'$ is safe, i.e., that it satisfies $\sum_a \pi'_a c_a \leq 0$. To alleviate this uncertainty, we construct a hybrid arm-selection policy $\boldsymbol{\pi}_t$ that adaptively

blends exploration with safety-aware exploitation. Specifically, the policy at time $t$ is defined as

$$\pi_t(i) = q_t \cdot \pi'(i) + (1 - q_t) \cdot \frac{e^{-\zeta_t \hat{C}_{t-1}(i)}}{\sum_{j \in [K]} e^{-\zeta_t \hat{C}_{t-1}(j)}}, \quad (3)$$

where $\hat{C}_t(i) := \sum_{s=1}^{t} \frac{Z_s(i)}{\pi_s(i)} \mathbb{I}\{a_s = i\}$ serves as the inverse propensity weighting estimator of the cumulative cost up to time $t$, and $q_t, \zeta_t$ are time-varying hyperparameters. The second term in (3) is a Gibbs-based policy (Sutton & Barto, 2018), which is a standard construction for mapping cost estimates to a risk-averse probability distribution. This structure ensures that actions with higher predicted costs are exponentially penalized, thereby providing a safety buffer during the exploratory phase. In Theorem 4.1, we establish an upper bound on the estimation error of the proposed algorithm.

**Theorem 4.1** (Statistical Estimation Error Upper Bound). *Let $\hat{\theta}$ denote the estimator produced by Phase I of the SER-MiSC algorithm in Algorithm 1, with the hyperparameter sequence $\{q_t\}_{t \geq 1}$ chosen such that $\underline{q} := \inf_{t \geq 1} q_t > 0$ is a constant independent of $T_1$. Then, $\exists \underline{t}_0 \in \mathbb{N}$ such that for any $T_1 \geq \underline{t}_0$, it holds that*

$$\mathbb{E}\big[||\hat{\theta} - \theta^*||_2\big] \leq \sqrt{\frac{11\sigma^2}{9\underline{q}T_1} tr\left(\left(\sum_{i=1}^{K} \pi'(i)x_i x_i^\top\right)^{-1}\right)}$$
$$= O(T_1^{-\frac{1}{2}}).$$

Before presenting the theoretical safety analysis for Phase I, we introduce the following assumption regarding the costs.

**Assumption 4.2** (Bounded Safety Costs). *For all $t \in [T]$ and $a \in [K]$, we assume the expected cost $c_a = \mathbb{E}[Z_t(a)]$ and the realized cost $Z_t(a)$ satisfy the following conditions:*

1. **Expected Cost Bounds**: *There exists a constant $\bar{c} > 0$ such that $\max_{a \in [K]} c_a < \bar{c}$. Furthermore, let $c_{\min} = \min_{a \in [K]} c_a$; there exists a constant $\varsigma > 0$ such that $|c_{\min}| \geq \varsigma \bar{c}$.*

2. **Almost Sure Bound**: *The realized costs are almost surely bounded, i.e., there exists a constant $\bar{Z} > 0$ such that $Z_t(a) \leq \bar{Z}$ for all $a \in [K]$ and $t \in [T]$.*

The almost sure boundedness of $Z_t(a)$ is standard in the constrained bandit literature (Liu et al., 2021; Ma et al., 2024). The non-degeneracy condition $|c_{\min}| \geq \varsigma \bar{c}$ ensures that at least one arm provides a meaningful safety buffer, preventing degenerate cases where all arms incur non-negative costs and safe exploration becomes impossible.

By setting

$$\zeta_t = D\left(\max_{j \in [K]} B(j) \sqrt{\mathbb{I}\{t = 1\} + \sum_{s=1}^{t-1} \frac{1}{q_s}}\right)^{-1}, \forall t \in [T_1] \quad (4)$$

in Algorithm 1 with $B(i)$ defined as:

$$B(i) = 2(\bar{Z} + 2\bar{c})\left(\frac{1}{\pi'(i)} + \frac{1}{\pi'(i_{\min})}\right)\sqrt{q_1}/\underline{q}, \forall i \in [K]$$

where $D > 0$ is a tuning hyperparameter, we obtain the following upper bound on the expected safety violation during Phase I.

**Theorem 4.3** (Safety Constraint Violation Upper Bound for Phase I). *Let $\mathcal{C}_1(\tau)$ be the safety cost incurred during Phase I of the SERMiSC algorithm in Algorithm 1 up to time $\tau \leq T_1$. We define the additional safety cost as*

$$\Delta\mathcal{C}_1(\tau) := \mathcal{C}_1(\tau) - \left[\sum_{i=1}^{K} c_i \pi'(i) \sum_{t=1}^{\tau} q_t + c_{\min} \sum_{t=1}^{\tau}(1 - q_t)\right].$$

*Set $\zeta_t$ as in (4). Then, under Assumption 4.2, $\exists \underline{t}_1 \in \mathbb{N}$ such that for any $T_1 \geq \underline{t}_1$, it holds that*

$$\Delta\mathcal{C}_1(T_1) \leq 2 \sum_{i \neq i_{\min} \wedge c_i > 0} c_i \left[2K \sum_{t=1}^{T_1}(1 - q_t)\right]^{\frac{D}{D+1}}$$
$$\times \left[(1 - \underline{q})(1 + \frac{2}{W(i)^2 D^2})\right]^{\frac{1}{D+1}},$$

*where $W$ represents a set of constants independent of $T_1$, and $D$ is a hyperparameter. Furthermore, by choosing $D = 1/\ln T_1$, we obtain the following bound on the additional safety cost:*

$$\Delta\mathcal{C}_1(T_1) = O\big((\ln T_1)^2\big).$$

The following corollary instantiates the safety bound from Theorem 4.3 by considering a uniform exploration strategy, thereby providing a clearer characterization of the algorithm's behavior.

**Corollary 4.4.** *Suppose Assumption 4.2 holds. By setting $q_t = \frac{\varsigma}{1+\varsigma}$ for all $t \in [T_1]$ and letting $\pi'$ be the uniform distribution (i.e., $\pi' = \frac{1}{K}\mathbf{1}$), the expected safety cost satisfies: $\mathcal{C}_1(T_1) \leq \Delta\mathcal{C}_1(T_1) = O\big((\ln T_1)^2\big)$.*

*Proof of Corollary 4.4.* It suffices to verify the fact that $\sum_{i=1}^{K} c_i \pi'(i) \sum_{t=1}^{T_1} q_t + c_{\min} \sum_{t=1}^{T_1}(1 - q_t) \leq \frac{T_1}{\varsigma+1} \frac{1}{K} \sum_{i=1}^{K} (\varsigma c_i - |c_{\min}|) \leq 0.$ $\qquad\square$

Detailed proofs for all results in Section 4.1 are provided in Appendix C.

*Remark* 4.5 (On the Choice of the Exploratory Policy). The uniform policy in Corollary 4.4 spreads the sampling budget evenly across all arms, so the constant factor in the estimation error bound of Theorem 4.1 is not yet optimized. A constrained optimal experimental design could instead allocate the budget more efficiently across arms, improving this constant factor. Notably, under the mixing schedule of Corollary 4.4, the safety guarantee holds for any lower-bounded exploration policy and requires no knowledge of the cost vector, so the optimal design depends only on the known feature vectors and can be computed offline.

### 4.2. Phase II: Safe Regret Minimization

Let $T_2$ denote the temporal length of Phase II. The details of the SERMiSC algorithm during Phase II are presented in Algorithm 2. The design of our arm selection strategy:

$$a_t = \arg\max_{i \in [K]} \hat{r}_t(i) - \frac{Z_t(i)Q(t)}{V_t},$$

where $\hat{r}_t(i)$ is an optimistic estimation of arm $i$'s expected reward, addresses the core challenge where the expected costs are unknown from the very beginning, rendering the baseline policy $\boldsymbol{\pi}^*$ inaccessible.

To effectively navigate the trade-off between maximizing rewards and ensuring safety, we adopt a pessimistic-optimistic framework (Liu et al., 2021). Central to this design is the dynamic weight $Q(t)$, which functions as a virtual queue explicitly tracking the level of safety constraint violation. The magnitude of $Q(t)$ serves as a direct feedback mechanism: a severe safety violation results in a large $Q(t)$, which amplifies the penalty for risky actions and compels the agent to focus on selecting more conservative arms to alleviate the safety violation.

To theoretically justify this discrete selection mechanism, we employ Lyapunov drift analysis. This analytical framework allows us to control the deviation of the algorithm's actual discrete choices $\mathbb{I}\{a_t = a\}$ from a reference mixed strategy $\boldsymbol{\pi}$ (typically related to, but not necessarily identical to, the baseline strategy $\boldsymbol{\pi}^*$). Specifically, as we will show in Appendix D, the relationship between the algorithm's choice and the reference strategy is governed by the drift of the Lyapunov function $L(t) := \frac{1}{2}Q(t)^2$:

$$\sum_{j \in [K]} \hat{r}_t(j)\Big[\pi_j - \mathbb{I}\{a_t = j\}\Big]$$
$$\leq \frac{Q(t)}{V_t}\Big[\sum_{j \in [K]} Z_t(j)\pi_j + \epsilon_t\Big] - \frac{L(t+1) - L(t)}{V_t}.$$

Based on the aforementioned approach, we derive the upper bound for the cumulative regret in Theorem 4.7 under the

---

**Algorithm 2** SERMiSC - **Phase II:** Regret Minimization

1: **Input:** $V_t, \epsilon_t, \varrho, \lambda$
2: $Q(1 + T_1) \leftarrow 0$
3: $\Sigma \leftarrow \lambda I_d$
4: $C_{T_1}(\varrho) \leftarrow \{\theta \in \mathbb{R}^d : ||\theta||_\Sigma \leq \sqrt{\lambda}S + \sigma\sqrt{d\log(1/\varrho)}\}$
5: **for** $t = 1 + T_1, ..., T$ **do**
6:  $\quad \hat{r}_t(i) \leftarrow \text{clip}_{[-1,1]}\Big[\max_{\theta \in C_{t-1}(\varrho)}\langle\theta, x_i\rangle\Big], \forall i \in [K]$
7:  $\quad$ pull the arm

$$a_t \leftarrow \arg\max_{i \in [K]} \hat{r}_t(i) - \frac{Z_t(i)Q(t)}{V_t}$$

8:  $\quad$ observe the reward $y_t$
9:  $\quad$ compute

$$Q(t+1) \leftarrow \Big[Q(t) + \sum_{i \in [K]} Z_t(i)\mathbb{I}\{a_t = i\} + \epsilon_t\Big]^+$$

10: $\quad \Sigma \leftarrow \Sigma + x_{a_t}x_{a_t}^\top$
11: $\quad \hat{\theta} \leftarrow \Sigma^{-1}\sum_{s=1+T_1}^{t} y_{a_s}x_{a_s}$
12: $\quad C_t(\varrho) \quad \leftarrow \quad \Big\{\theta \in \mathbb{R}^d : ||\hat{\theta} - \theta||_\Sigma \leq \sqrt{\lambda}S + \sigma\sqrt{d\log\Big(\frac{1+(t-T_1)L^2/\lambda}{\varrho}\Big)}\Big\}$
13: **end for**

---

following settings and assumptions. First, we consider the parameters and features to be bounded. Let $M$ be a uniform upper bound for the expected rewards such that $|\mu_j| \leq M$ for all $j \in [K]$; for notational simplicity, we assume $M = 1$. Similarly, we assume that the true parameter vector $\theta^*$ and the feature vectors $x_j$ are uniformly bounded, such that $\|\theta^*\|_2 \leq S$ and $\|x_j\|_2 \leq L$ for all $j \in [K]$. To ensure the existence of at least one safe strategy, we require the following Slater's Assumption:

**Assumption 4.6** (Slater's Assumption (Liu et al., 2021)). There exists a constant $\delta \in (0, 1)$ such that $\mathcal{P}(\delta)$ is feasible, and let $\boldsymbol{\pi}_\delta$ be a policy in its feasible set.

Slater's condition is a standard constraint qualification in constrained optimization (Liu et al., 2021). It requires the existence of a policy $\boldsymbol{\pi}_\delta$ that strictly satisfies the safety constraint with a positive margin $\delta$, i.e., $\langle\boldsymbol{\pi}_\delta, \boldsymbol{c}\rangle \leq -\delta < 0$. This strict feasibility provides the safety buffer that drives the virtual queue $Q(t)$ downward in the Lyapunov drift analysis, and the margin $\delta$ will appear explicitly in our regret bounds. Under these conditions, we establish the regret upper bound for Phase II as follows:

**Theorem 4.7** (Regret Upper Bound for Phase II). *Let $\mathcal{R}_2(\tau)$ be the cumulative regret incurred during Phase II of the SERMiSC algorithm in Algorithm 2 over the period from time $1 + T_1$ to $\tau \leq T$. Under Assumption 4.6, by setting*

$V_t = \delta\sqrt{t}$, $\epsilon_t = 4/\sqrt{t}$, and $\varrho = 1/T$, it holds that

$$\mathcal{R}_2(T_2) \leq 4 + 16\left[\frac{3}{\delta} + \frac{1}{\delta^2} + \frac{4(\bar{Z}+4)}{\delta^3}\right]$$

$$+ \frac{2(\bar{Z}^2+4)}{\delta}\sqrt{T_2} + 2\left(\sigma\sqrt{d\log\left[T_2\left(1 + \frac{T_2 L^2}{\lambda}\right)\right]}\right)$$

$$+ \sqrt{\lambda}S\right)\sqrt{2T_2\log\left(1 + \frac{T_2 L^2}{\lambda d}\right)},$$

which implies an asymptotic upper bound of $\mathcal{R}_2(T_2) = \tilde{O}\left(\frac{\sqrt{T}}{\delta} + \frac{1}{\delta^3}\right)$.

We also establish an upper bound on the safety constraint violation in the following theorem.

**Theorem 4.8** (Safety Constraint Violation Upper Bound for Phase II). *Let $\mathcal{C}_2(\tau)$ be the safety cost incurred during Phase II of the SERMiSC algorithm in Algorithm 2 over the period from time $1 + T_1$ to $\tau \leq T$. Under Assumption 4.6, by setting $V_t = \delta\sqrt{t}$, $\epsilon_t = 4/\sqrt{t}$, and $\varrho = 1/T$, it holds that*

$$\mathcal{C}_2(T_2) \leq 8 + \frac{(\bar{Z}+4)^2 + (\bar{Z}+4)\delta/24}{\delta/8}$$

$$\times \log\left(1 + \frac{16(\bar{Z}+4)^2 + 2(\bar{Z}+4)\delta/3}{\delta^2/8}\right)$$

$$+ \left(1 + \lceil\frac{64}{\delta^2}\rceil\right)(\bar{Z}+4) + \frac{4(\bar{Z}^2+16)}{\delta},$$

*implying that the safety cost remains $O(1)$ relative to $T_2$.*

Detailed proofs for all results in Section 4.2 are provided in Appendix D.

### 4.3. Pareto Optimality of the Proposed Algorithm

In this subsection, we demonstrate the Pareto optimality of the SERMiSC algorithm with respect to cumulative regret, safety violation, and estimation error. The theoretical analysis for each phase of the algorithm is summarized in Table 1. We introduce a hyperparameter $\beta \in [1/2, 1)$ such that $T_1 = \lfloor T^\beta \rfloor$. This parameter characterizes the relative duration of Phase I and Phase II, thereby quantifying the fundamental trade-off between estimation error and cumulative regret under safety constraints.

**Corollary 4.9** (End-to-End Performance Guarantees of SERMiSC). *Let $\beta \in [1/2, 1)$, $T_1 = \lfloor T^\beta \rfloor$, and $T_2 = T - T_1$. Under Assumptions 4.2 and 4.6, with the parameter choices specified in Section 4, the SERMiSC algorithm simultaneously achieves:*

$$\text{(Regret)} \quad \mathcal{R}(T) = \tilde{O}(T_1 + \sqrt{T_2}) = \tilde{O}(T^\beta),$$

$$\text{(Safety)} \quad \mathcal{C}(T) = \tilde{O}(1),$$

$$\text{(Estimation)} \quad \text{err}(T) = O\left(T^{-\beta/2}\right).$$

These bounds follow directly from summing the per-phase guarantees in Table 1: the cumulative regret is $\tilde{O}(T_1 + \sqrt{T_2}) = \tilde{O}(T^{\max\{\beta, 1/2\}}) = \tilde{O}(T^\beta)$, while the total safety violation remains $\tilde{O}(1)$ and the estimation error is $O(1/\sqrt{T^\beta})$. Consequently, $\pi_{\text{SERMiSC}} \in \Pi(\beta, \delta_T)$ with $\delta_T = \tilde{O}(1)$. Given that Theorem 3.1 establishes a minimax lower bound on the estimation error of $\Omega(1/\sqrt{T^\beta})$ for this class, SERMiSC achieves the optimal rate up to logarithmic factors. This effectively demonstrates the Pareto optimality of our proposed algorithm.

*Remark* 4.10. While setting $\beta < 1/2$ (i.e., $T_1 = o(T^{1/2})$) is technically feasible, this regime is strictly dominated by the choice of $\beta = 1/2$. Specifically, for $\beta < 1/2$, the algorithm incurs a strictly larger estimation error without yielding any reciprocal improvement in either cumulative regret or safety performance. Consequently, our discussion focuses on the rational regime where $\beta \in [1/2, 1)$.

## 5. Simulations

In this section, we provide numerical simulations to evaluate the performance of the proposed SERMiSC algorithm across three key metrics: cumulative regret, safety constraint violation, and statistical estimation error.

**Experimental Setup.** We consider a linear bandit environment with time horizon $T = 1000$, $K = 10$ arms, and feature dimension $d = 8$. The arm set consists of the eight standard basis vectors $e_1, \ldots, e_8$ together with two dense vectors $(0.1, \ldots, 0.1)^\top$ and $(0.1, -0.1, \ldots, -0.1)^\top$, providing a mix of sparse and correlated actions. The hidden parameter is $\theta^* = (1, 5, -3, 0.2, 3, 3, -4, 9)^\top$ and the mean safety cost vector is $c = (2, -2, -5, 0, 0, 0, 0, 4, 1, 1)^\top$, so that several arms carry positive expected cost and a safe policy must randomize non-trivially. Reward and cost noise are i.i.d. Gaussian with standard deviations $\sigma = 0.5$ and $\sigma_c = 0.05$, respectively. For SERMiSC, the Phase I exploration policy $\pi'$ is uniform over arms, the mixing coefficient is $q_t \equiv 0.5$, and the ridge regularization is $\lambda = 0.1$. All results are averaged over 10 independent trials.

**Experiment 1: Impact of Phase I Duration ($T_1$).** As illustrated in Figure 1, we investigate how the duration of Phase I, $T_1 \in \{70, 200, 400, 700\}$, influences the algorithm's performance. The results reveal a clear trade-off: an increase in $T_1$ leads to higher cumulative regret and safety violations; however, it significantly enhances estimation accuracy. This empirical observation aligns with the theoretical upper bounds summarized in Table 1, validating the consistency of our analysis. In practice, $T_1$ should be chosen to reflect the relative priority of inference versus online reward: a larger $T_1$ favors estimation accuracy at the cost of cumulative regret, while a smaller $T_1$ prioritizes reward performance.

**Experiment 2: Comparative Analysis with Baselines.** We

*Table 1.* Theoretical Performance Guarantees of the SERMiSC Algorithm.

| METRIC | PHASE I | PHASE II | TOTAL |
|---|---|---|---|
| REGRET | $\Theta(T_1)$ | $\tilde{O}(\sqrt{T_2})$ (THEOREM 4.7) | $\tilde{O}(T_1 + \sqrt{T_2})$ |
| SAFETY VIOLATION | $\tilde{O}(1)$ (COROLLARY 4.4) | $O(1)$ (THEOREM 4.8) | $\tilde{O}(1)$ |
| ESTIMATION ERROR | $O(1/\sqrt{T_1})$ (THEOREM 4.1) | N/A | $O(1/\sqrt{T_1})$ |

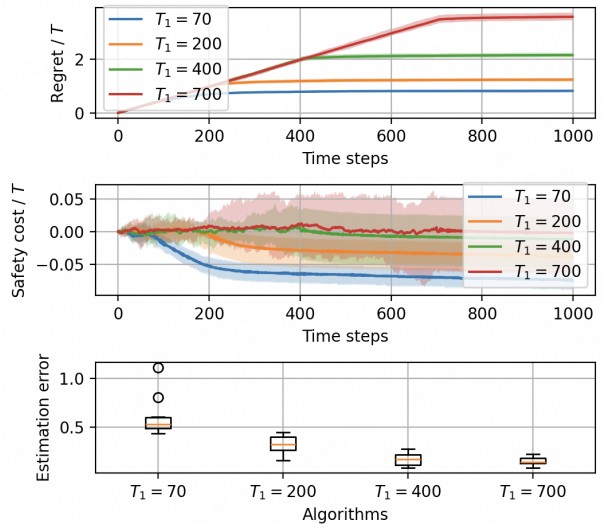

*Figure 1.* Performance of the proposed SERMiSC algorithm under various Phase I durations $T_1 \in \{70, 200, 400, 700\}$.

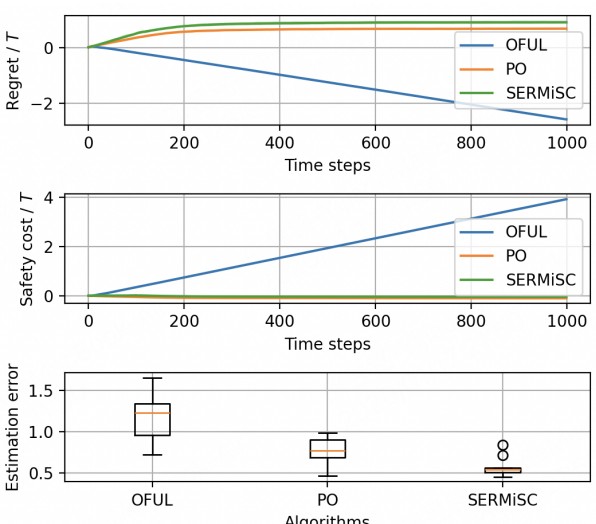

*Figure 2.* Performance comparison: SERMiSC vs. OFUL (Abbasi-Yadkori et al., 2011) and PO (Liu et al., 2021).

benchmark SERMiSC (with $T_1 = 100$) against two representative algorithms: OFUL (Abbasi-Yadkori et al., 2011) and PO (Liu et al., 2021). We observe that OFUL incurs divergent safety violations. In contrast, while achieving regret and safety performance comparable to PO, SERMiSC significantly outperforms PO in terms of estimation error. This demonstrates the efficiency of our proposed two-phase structure in obtaining high-fidelity parameter estimates without compromising reward and safety.

## 6. Conclusion

This paper investigates the fundamental trade-off between regret and statistical inference under safety constraints in linear stochastic bandits. We characterize the Pareto frontier by deriving a lower bound on estimation error under specific regret and safety budgets, a bound which is matched by our novel algorithm. These results provide new insights into the design of safety-aware, multi-objective decision-making.

**Future Work.** Two directions merit further investigation. First, our analysis assumes a linear reward model, which offers analytical clarity but may not faithfully capture real-world reward behavior. Extending the framework to richer settings, such as heteroscedastic noise, nonlinear reward

models, or bounded model misspecification, is a natural next step, and we expect the two-phase design principle to remain applicable once the confidence and estimation machinery are reformulated accordingly. Second, SERMiSC adopts an explicit two-phase split between estimation and regret minimization, which causes the regret to accumulate linearly during the estimation phase. Designing a fully adaptive algorithm that interleaves estimation and regret minimization, smoothing out this early linear growth while still attaining the same Pareto-optimal trade-off, is a promising direction for future work.

## Acknowledgements

The authors would like to thank Zhiheng Zhang for insightful discussions. This work was supported by Xiongan AI Institute and Tsinghua University Dushi Program.

## Impact Statement

This paper presents work whose goal is to advance the field of Machine Learning. There are many potential societal consequences of our work, none which we feel must be specifically highlighted here.

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

# A. Extended Related Works

## A.1. The Trade-off between Regret and Statistical Inference

While the conventional multi-armed bandit (MAB) and linear stochastic bandit literature focus primarily on either regret minimization (Abbasi-Yadkori et al., 2011; Ghosh et al., 2017; Hao et al., 2020; Kveton et al., 2020; Charisopoulos et al., 2023) or pure exploration tasks such as best arm identification (Soare et al., 2014; Tao et al., 2018; Degenne et al., 2020; Zaki et al., 2022; Shao & Fang, 2025), a growing body of recent work explores the fundamental tension between these two objectives. To the best of our knowledge, the seminal study by Simchi-Levi & Wang (2025) first formally characterized this trade-off, demonstrating that the objectives of minimizing cumulative regret and maximizing statistical estimation precision—specifically, minimizing the expected error of the Average Treatment Effect (ATE) estimator—are inherently in conflict. This necessitates a characterization of the Pareto-optimal frontier to navigate the balance between estimation efficiency and cumulative performance. Building upon this foundation, several recent studies have extended the regret-inference trade-off to various more complex structural and environmental settings:

- *Non-stationary Environments.* Simchi-Levi et al. (2023) extended the trade-off to experimental designs characterized by non-stationary rewards following linear trends. Unlike the standard linear stochastic bandit setting where the underlying parameters are typically assumed to be fixed, their framework treats the expected reward as a time-evolving process with unknown drift coefficients. Consequently, their work can be viewed as an online decision-making problem where covariates are strictly time-dependent and predictable, and the objective includes the precise estimation of these drift parameters alongside regret minimization.

- *Network Interference.* In scenarios where a unit's reward is simultaneously influenced by the treatment assignments of other units, Zhang & Wang (2025) and Wang et al. (2025) investigated how such interference complicates the trade-off within online experimental design. A distinct feature of these works compared to ours also lies in their estimation target: they primarily focus on the ATE within a potential outcomes framework (Rubin, 1974), following the non-parametric tradition of Simchi-Levi & Wang (2025). These models often operate without a global parametric structure, focusing instead on identifying causal contrasts between treatments. In contrast, our linear stochastic bandit setting leverages a parametric model-based approach, where the objective is the precise estimation of the underlying parameter vector $\theta^*$.

- *High-Dimensional Settings.* Duan et al. (2024) investigated the interplay between regret and estimation within high-dimensional sparse linear bandits. Notably, they demonstrated that the Pareto conflict can be mitigated when the environment satisfies a covariate diversity condition. This condition, a standard assumption in high-dimensional bandit theory, requires the population design matrix of the arm covariates to have a minimum eigenvalue strictly bounded away from zero. Under this assumption, the actions selected for regret minimization naturally provide sufficient information to identify the parameter vector $\theta^*$. Consequently, satisfactory estimation performance can be achieved as a by-product of exploitation, without the need for additional estimation effort.

## A.2. Safety-aware Linear Bandits

Another line of research directly relevant to our work is safety-aware linear bandits, where the objective is to minimize cumulative regret while ensuring the agent's actions satisfy specific safety requirements. The existing body of research in this area highlights two critical insights: first, ensuring safety has become an indispensable and increasingly vital requirement in linear bandit applications; second, despite this growing importance, these studies have primarily focused on regret minimization and have largely neglected the objective of statistical estimation. While the literature has established various robust mechanisms for safe exploration, it offers several distinct formulations of safety constraints:

- *High-probability Safety Constraints.* A representative example within the safety-aware literature is the enforcement of stringent anytime high-probability constraints. Amani et al. (2019) pioneered this direction by requiring that every action taken by the learner remains within a safe region with high probability. Their algorithm, starting from a known initial seed set of safe arms, employs a conservative expansion strategy: it only explores actions whose safety can be statistically certified based on current observations. By strictly confining exploration to this sequentially expanding empirical boundary, the framework guarantees zero safety violations throughout the entire horizon with high probability. This methodology was later generalized to non-linear reward structures by Amani et al. (2020) in the context of generalized linear bandits.

- *Conservativeness-style Safety Constraints.* Kazerouni et al. (2017) pioneered the conservative bandit model, which focuses on a cumulative performance constraint. This approach ensures that the learner's total expected reward remains above a fixed percentage of a baseline policy's performance at any time $t$. In contrast, Khezeli & Bitar (2020) introduced a more stringent stagewise safety constraint, requiring the learner's instantaneous expected reward to stay above a prescribed threshold $b$ at every single stage. Crucially, this requirement is formulated as a PAC-style guarantee governed by a risk tolerance parameter $\delta \in [0, 1]$, which explicitly quantifies the maximum allowable probability of safety violation at each step. By allowing for this controlled risk, the learner can systematically expand its set of safe actions while maintaining high-probability reliability—a feature particularly vital in high-stakes applications like clinical trials where step-by-step operational safety is paramount.

- *Expectation-based Safety Constraints.* Most similar to our work are the primal-dual frameworks provided by Liu et al. (2021) and Ma et al. (2024), which formulate safety constraints in expectation. This formulation functions as a relaxation of high-probability constraints, prioritizing average performance across trials rather than worst-case realizations. Liu et al. (2021) leverage this relaxation to develop an efficient pessimistic-optimistic algorithm for general constraints (encompassing safety, fairness, budget limits, among others). In parallel, Ma et al. (2024) address the high-dimensional contextual bandits with knapsacks (CBwK) problem. They develop an online hard thresholding (Online HT) estimator to harness sparsity directly within a primal-dual scheme. This integrated approach allows them to update dual variables via online learning rules, achieving a sub-linear regret that scales only logarithmically with the feature dimension, thereby overcoming the polynomial dependency found in prior work. Both works exemplify how Lagrangian-based techniques can effectively manage the feasibility-optimality trade-off in linear bandit scenarios.

## B. Pareto Optimal Trade-off

In this part, we derive the minimax lower bound for statistical estimation power under the requirements for both regret and constraint violation in order to demonstrate the trade-off among these objectives. For maximum generality, we establish the bound under the *anytime* safety formulation

$$\Pi^{\text{any}}(\beta, \boldsymbol{\delta}) := \left\{ \boldsymbol{\pi} \quad s.t. \quad \begin{cases} \mathcal{R}(T) = O(T^{\beta}) \\ \mathcal{C}(t) = O(\delta_t), \ \forall t \in [T] \end{cases} \right\},$$

for an arbitrary tolerance sequence $\boldsymbol{\delta} = \{\delta_t\}_{t \in [T]}$ with terminal value $\delta_T$. Since $\Pi^{\text{any}}(\beta, \boldsymbol{\delta}) \subseteq \Pi(\beta, \delta_T)$ (imposing anytime constraints only shrinks the admissible class), the statement of Theorem 3.1 follows immediately from the bound we prove below. To achieve this, we need to construct hard instances as follows. There are two environments $\nu_1, \nu_2$ that are difficult to distinguish. We set

$$\theta_1^* = \begin{bmatrix} 1 \\ 0 \end{bmatrix}, \quad \theta_2^* = \begin{bmatrix} 1 \\ 2\Delta \end{bmatrix},$$

correspond to the environments $\nu_1, \nu_2$, respectively, where $\Delta > 0$ is a constant. The variance of the stochastic noise in both environments is $\sigma'^2$. As for the arm set, we propose constructing three arms. We need $\mu_2 > \mu_1 > \mu_3$, $c_1 < 0$, and $c_2, c_3 > 0$. The following construction satisfies these requirements:

$$x_1 = \begin{bmatrix} 0.2 \\ 0 \end{bmatrix}, \quad x_2 = \begin{bmatrix} 0.4 \\ 0 \end{bmatrix}, \quad x_3 = \begin{bmatrix} 0.1 \\ 0.1 \end{bmatrix}, \quad c_1 = -1, \quad c_2 = 1, \quad c_3 = 2.$$

To derive a lower bound for the estimation error, we need to show that, for whatever estimator we use, there exists a problem instance such that the probability $\Pr\left( ||\hat{\theta} - \theta^*||_2 \geq \Delta \right)$ is non-negligible. By our problem instance construction, it suffices to solve a simple two hypothesis testing problem. Define a minimum distance test $\psi(\hat{\theta}) = \underset{i \in \{1,2\}}{\arg\min} \, ||\hat{\theta} - \theta_i^*||_2$. By the triangle inequality,

$$\begin{aligned} &||\theta_1^* - \theta_2^*||_2 \\ =&||\hat{\theta} - \theta_1^* - (\hat{\theta} - \theta_2^*)||_2 \\ \leq&||\hat{\theta} - \theta_1^*||_2 + ||\hat{\theta} - \theta_2^*||_2. \end{aligned}$$

Thus, the event $\psi(\hat{\theta}) = 1$ implies that

$$
\begin{aligned}
&||\hat{\theta} - \theta_2^*||_2 \\
\geq& \frac{1}{2}\left(||\hat{\theta} - \theta_1^*||_2 + ||\hat{\theta} - \theta_2^*||_2\right) \\
\geq& \frac{1}{2}||\theta_1^* - \theta_2^*||_2 \\
=& \frac{1}{2}\sqrt{0^2 + (-2\Delta)^2} = \Delta
\end{aligned}
$$

since $||\hat{\theta} - \theta_1^*||_2 \leq ||\hat{\theta} - \theta_2^*||_2$. Similarly, the event $\psi(\hat{\theta}) = 2$ implies that $||\hat{\theta} - \theta_1^*||_2 \geq \Delta$. We now have

$$
\begin{aligned}
&\inf_{\substack{\boldsymbol{\pi} \in \Pi^{\mathrm{any}}(\beta, \boldsymbol{\delta}) \\ \hat{\theta} \in \widehat{\Theta}}} \sup_{\nu \in \mathcal{V}} \mathbb{P}_\nu^{\boldsymbol{\pi}}\left(||\hat{\theta} - \theta_\nu^*||_2 \geq \Delta\right) \\
\geq& \inf_{\substack{\boldsymbol{\pi} \in \Pi^{\mathrm{any}}(\beta, \boldsymbol{\delta}) \\ \hat{\theta} \in \widehat{\Theta}}} \max_{i \in \{1,2\}} \mathbb{P}_{\nu_i}^{\boldsymbol{\pi}}\left(||\hat{\theta} - \theta_i^*||_2 \geq \Delta\right) \\
\geq& \inf_{\substack{\boldsymbol{\pi} \in \Pi^{\mathrm{any}}(\beta, \boldsymbol{\delta}) \\ \hat{\theta} \in \widehat{\Theta}}} \max_{i \in \{1,2\}} \mathbb{P}_{\nu_i}^{\boldsymbol{\pi}}(\psi(\hat{\theta}) \neq i) \\
\geq& \frac{1}{2} \inf_{\substack{\boldsymbol{\pi} \in \Pi^{\mathrm{any}}(\beta, \boldsymbol{\delta}) \\ \hat{\theta} \in \widehat{\Theta}}} \left[\mathbb{P}_{\nu_1}^{\boldsymbol{\pi}}(\psi(\hat{\theta}) \neq 1) + \mathbb{P}_{\nu_2}^{\boldsymbol{\pi}}(\psi(\hat{\theta}) \neq 2)\right] \\
\geq& \frac{1}{2} \inf_{\substack{(\boldsymbol{\pi}, \tilde{\psi}): \\ \boldsymbol{\pi} \in \Pi^{\mathrm{any}}(\beta, \boldsymbol{\delta})}} \left[\mathbb{P}_{\nu_1}^{\boldsymbol{\pi}}(\tilde{\psi} \neq 1) + \mathbb{P}_{\nu_2}^{\boldsymbol{\pi}}(\tilde{\psi} \neq 2)\right] \\
=& \frac{1}{2}\left[1 - \sup_{\boldsymbol{\pi} \in \Pi^{\mathrm{any}}(\beta, \boldsymbol{\delta})} \mathrm{TV}(\mathbb{P}_{\nu_1}^{\boldsymbol{\pi}}, \mathbb{P}_{\nu_2}^{\boldsymbol{\pi}})\right] \\
\geq& \frac{1}{2}\left[1 - \sqrt{\frac{1}{2} \sup_{\boldsymbol{\pi} \in \Pi^{\mathrm{any}}(\beta, \boldsymbol{\delta})} \mathrm{KL}(\mathbb{P}_{\nu_1}^{\boldsymbol{\pi}} || \mathbb{P}_{\nu_2}^{\boldsymbol{\pi}})}\right].
\end{aligned}
$$

The second inequality is due to our argument that the event $\psi(\hat{\theta}) \neq i$ implies that $||\hat{\theta} - \theta_i^*||_2 \geq \Delta$. The fourth inequality follows from relaxing the infimum from tests of the form $\psi(\hat{\theta})$ to all measurable tests $\tilde{\psi}$ mapping the observation history $(a_1, y_1, \ldots, a_T, y_T)$ to $\{1, 2\}$; since tests of the form $\psi(\hat{\theta})$ are a special case of $\tilde{\psi}$, taking the infimum over the larger class can only decrease the value. The equality is derived from the Neyman-Pearson Lemma, and the last inequality is due to Pinsker's inequality. Consequently, lower bounding the minimax error probability suffices to upper bound $\mathrm{KL}(\mathbb{P}_{\nu_1}^{\boldsymbol{\pi}} || \mathbb{P}_{\nu_2}^{\boldsymbol{\pi}})$. By the same argument in (Lattimore & Szepesvári, 2020), it can be decomposed such that $\mathrm{KL}(\mathbb{P}_{\nu_1}^{\boldsymbol{\pi}} || \mathbb{P}_{\nu_2}^{\boldsymbol{\pi}}) = \sum_{t=1}^{T} \sum_{a \in [K]} \mathrm{KL}_a(P_{\nu_1,t}^{\boldsymbol{\pi}} || P_{\nu_2,t}^{\boldsymbol{\pi}}) P_{\nu_1,t}^{\boldsymbol{\pi}}(a_t = a)$, where $P_{\nu,t}^{\boldsymbol{\pi}}$ is the probability measure with respect to time slot $t$.

Since we require that our algorithm's arm selection strategy $\boldsymbol{\pi}$ satisfies both regret and constraint violation requirements, the upper bound of $\mathrm{KL}(\mathbb{P}_{\nu_1}^{\boldsymbol{\pi}} || \mathbb{P}_{\nu_2}^{\boldsymbol{\pi}})$ is determined by the following linear program:

$$
\max_{\{P_{\nu_1,t}^{\boldsymbol{\pi}}(a_t = a)\}_{t \in [T], a \in [K]}} \sum_{t=1}^{T} \sum_{a \in [K]} \mathrm{KL}_a(P_{\nu_1,t}^{\boldsymbol{\pi}} || P_{\nu_2,t}^{\boldsymbol{\pi}}) P_{\nu_1,t}^{\boldsymbol{\pi}}(a_t = a),
$$

$$
\begin{aligned}
s.t. \quad & \sum_{i=1}^{t} \sum_{a \in [K]} c_a P_{\nu_1,i}^{\boldsymbol{\pi}}(a_i = a) \leq \delta_t, \quad \forall t \in [T], \quad \text{(constraint violation)} \\
& T \sum_{a \in [K]} \pi_a^* \mu_a - \sum_{t=1}^{T} \sum_{a \in [K]} \mu_a P_{\nu_1,t}^{\boldsymbol{\pi}}(a_t = a) \leq CT^\beta, \quad \text{(regret)} \\
& \sum_{a \in [K]} P_{\nu_1,t}^{\boldsymbol{\pi}}(a_t = a) = 1, \quad \forall t \in [T], \\
& 0 \leq P_{\nu_1,t}^{\boldsymbol{\pi}}(a_t = a) \leq 1, \quad \forall t \in [T], a \in [K],
\end{aligned}
$$

where $\boldsymbol{\mu}, \boldsymbol{c}$, and $\boldsymbol{\pi}^*$ correspond to the environment $\nu_1$. By our instance construction, we have $\mathrm{KL}_1(P^{\boldsymbol{\pi}}_{\nu_1,t}\| P^{\boldsymbol{\pi}}_{\nu_2,t}) = \mathrm{KL}_2(P^{\boldsymbol{\pi}}_{\nu_1,t}\| P^{\boldsymbol{\pi}}_{\nu_2,t}) = 0$ and $\mathrm{KL}_3(P^{\boldsymbol{\pi}}_{\nu_1,t}\| P^{\boldsymbol{\pi}}_{\nu_2,t}) = \frac{0.02\Delta^2}{\sigma'^2}$. We can refine this linear program by replacing the variables $\{P^{\boldsymbol{\pi}}_{\nu_1,t}(a_t = a)\}_{t\in[T],a\in[K]}$ with $\{\mathbb{E}^{\boldsymbol{\pi}}_{\nu_1}[N_a(T)]\}_{a\in[K]}$, noting that $\mathbb{E}^{\boldsymbol{\pi}}_{\nu_1}[N_a(t)] = \sum_{s=1}^{t} P^{\boldsymbol{\pi}}_{\nu_1,s}(a_s = a)$. The optimal value of the above LP can be further upper bounded by the optimal value of another LP:

$$\max_{\{\mathbb{E}^{\boldsymbol{\pi}}_{\nu_1}[N_a(T)]\}_{a\in[K]}} \sum_{a\in[K]} \mathrm{KL}_a(P^{\boldsymbol{\pi}}_{\nu_1,T}\| P^{\boldsymbol{\pi}}_{\nu_2,T})\mathbb{E}^{\boldsymbol{\pi}}_{\nu_1}[N_a(T)],$$

$$s.t. \quad \sum_{a\in[K]} c_a \mathbb{E}^{\boldsymbol{\pi}}_{\nu_1}[N_a(T)] \leq \delta_T,$$

$$T\sum_{a\in[K]} \pi^*_a \mu_a - \sum_{a\in[K]} \mu_a \mathbb{E}^{\boldsymbol{\pi}}_{\nu_1}[N_a(T)] \leq CT^\beta,$$

$$\sum_{a\in[K]} \mathbb{E}^{\boldsymbol{\pi}}_{\nu_1}[N_a(T)] = T,$$

$$0 \leq \mathbb{E}^{\boldsymbol{\pi}}_{\nu_1}[N_a(T)] \leq T, \quad \forall a \in [K].$$

Note that we relaxed the anytime constraint to one that applies only with respect to time $T$. By defining variables $\tilde{\boldsymbol{\pi}}$ such that $\tilde{\pi}_a = \mathbb{E}^{\boldsymbol{\pi}}_{\nu_1}[N_a(T)]/T$, this linear program can be rewritten as

$$\max_{\tilde{\boldsymbol{\pi}}} \quad \sum_{a\in[K]} \mathrm{KL}_a(P^{\boldsymbol{\pi}}_{\nu_1,T}\| P^{\boldsymbol{\pi}}_{\nu_2,T})\tilde{\pi}_a T,$$

$$s.t. \quad \sum_{a\in[K]} c_a \tilde{\pi}_a T \leq \delta_T,$$

$$\sum_{a\in[K]} \mu_a(\pi^*_a - \tilde{\pi}_a)T \leq CT^\beta,$$

$$\tilde{\boldsymbol{\pi}} \in \Delta_K,$$

which is easier to analyze. We introduce a non-negative dual variable for each inequality constraint and a free dual variable for the equality constraint:

- $\lambda_1 \geq 0$ for Constraint Violation.

- $\lambda_2 \geq 0$ for Regret.

- $\iota \in \mathbb{R}$ for Sum to 1.

The Lagrangian $\mathcal{L}(\tilde{\boldsymbol{\pi}}, \boldsymbol{\lambda}, \iota)$ is formed by adding the objective function to the sum of the products of the dual variables and their respective constraints,

$$\mathcal{L}(\tilde{\boldsymbol{\pi}}, \boldsymbol{\lambda}, \iota) = \sum_{a\in[K]} \mathrm{KL}_a(P^{\boldsymbol{\pi}}_{\nu_1,T}\| P^{\boldsymbol{\pi}}_{\nu_2,T})\tilde{\pi}_a T$$

$$- \lambda_1\left(\sum_{a\in[K]} c_a T\tilde{\pi}_a - \delta_T\right) - \lambda_2\left(\sum_{a\in[K]} \mu_a(\pi^*_a - \tilde{\pi}_a)T - CT^\beta\right) - \iota\left(\sum_{a\in[K]} \tilde{\pi}_a - 1\right).$$

The dual objective function is $g(\boldsymbol{\lambda}, \iota) = \min_{\tilde{\boldsymbol{\pi}} \succeq 0} \mathcal{L}(\tilde{\boldsymbol{\pi}}, \boldsymbol{\lambda}, \iota)$. Since $\mathcal{L}$ is linear in $\tilde{\pi}_a$, the minimum with respect to $\tilde{\pi}_a$ is finite only if the coefficient of each $\tilde{\pi}_a$ is less than or equal to zero. To obtain the dual constraints, we rearrange the Lagrangian by grouping terms related to $\tilde{\pi}_a$:

$$\mathcal{L}(\tilde{\boldsymbol{\pi}}, \boldsymbol{\lambda}, \iota) = \sum_{a\in[K]} \tilde{\pi}_a\left[\mathrm{KL}_a(P^{\boldsymbol{\pi}}_{\nu_1,T}\| P^{\boldsymbol{\pi}}_{\nu_2,T})T - \lambda_1 c_a T - \lambda_2(-\mu_a T) - \iota\right]$$

$$+ \left[\lambda_1\delta_T + \lambda_2\left(CT^\beta - \sum_{a\in[K]} \mu_a\pi^*_a T\right) + \iota\right].$$

For the minimum to be non-trivial (i.e., not $-\infty$) when $\tilde{\pi}_a \geq 0$, the coefficient of each $\tilde{\pi}_a$ must be less than or equal to zero:

$$\mathrm{KL}_a(P^{\boldsymbol{\pi}}_{\nu_1,T} \| P^{\boldsymbol{\pi}}_{\nu_2,T})T - \lambda_1 c_a T + \lambda_2 \mu_a T \leq \iota, \quad \forall a \in [K].$$

As a consequence, the dual program can be written as

$$\min_{\boldsymbol{\lambda},\iota} \quad \lambda_1 \delta_T + \lambda_2 \left( CT^\beta - \sum_{a \in [K]} \mu_a \pi^*_a T \right) + \iota,$$

$$s.t. \quad \mathrm{KL}_a(P^{\boldsymbol{\pi}}_{\nu_1,T} \| P^{\boldsymbol{\pi}}_{\nu_2,T})T - \lambda_1 c_a T + \lambda_2 \mu_a T - \iota \leq 0, \quad \forall a \in \mathcal{K},$$

$$\boldsymbol{\lambda} \succeq 0.$$

Since $\iota$ is minimized in the objective and constrained to be greater than $K$ different values, the optimal $\iota$ will be equal to the maximum of those constraints:

$$\iota = \max_{a \in [K]} \left\{ \mathrm{KL}_a(P^{\boldsymbol{\pi}}_{\nu_1,T} \| P^{\boldsymbol{\pi}}_{\nu_2,T})T - \lambda_1 c_a T + \lambda_2 \mu_a T \right\}.$$

Substituting this $\iota$ into the objective yields the following simplified dual problem

$$\min_{\boldsymbol{\lambda} \succeq 0} \quad \lambda_1 \delta_T + \lambda_2 \left( CT^\beta - \sum_{a \in [K]} \mu_a \pi^*_a T \right) + \max_{a' \in [K]} \left\{ \mathrm{KL}_{a'}(P^{\boldsymbol{\pi}}_{\nu_1,T} \| P^{\boldsymbol{\pi}}_{\nu_2,T})T - \lambda_1 c_{a'} T + \lambda_2 \mu_{a'} T \right\}$$

$$= \min_{\boldsymbol{\lambda} \succeq 0} \quad \lambda_1 \delta_T + \lambda_2 CT^\beta + \max_{a' \in [K]} \left\{ \mathrm{KL}_{a'}(P^{\boldsymbol{\pi}}_{\nu_1,T} \| P^{\boldsymbol{\pi}}_{\nu_2,T})T - \lambda_1 c_{a'} T - \lambda_2 \left( \sum_{a \in [K]} \mu_a \pi^*_a - \mu_{a'} \right) T \right\}.$$

We investigate the properties of the strategy $\boldsymbol{\pi}^*$. Note that under the environment $\nu_1$, $\mu_1 = 0.2, \mu_2 = 0.4, \mu_3 = 0.1$. First, we must have $\pi^*_3 = 0$. Otherwise, an alternative strategy $\boldsymbol{\pi}$ such that $\pi_1 = \pi^*_1 + \pi^*_3$ and $\pi_2 = \pi^*_2$ would yield a lower expected cost and a higher expected reward, which contradicts the definition of $\boldsymbol{\pi}^*$. Second, it is also obvious that $\sum_{a \in [K]} \pi^*_a c_a = 0$. We have

$$\pi^*_1 c_1 + \pi^*_2 c_2 + \pi^*_3 c_3 = \pi^*_1 c_1 + \pi^*_2 c_2 = 0,$$

$$\pi^*_1 + \pi^*_2 = 1.$$

Thus, $\pi^*_1 = \frac{c_2}{c_2 - c_1}, \pi^*_2 = \frac{-c_1}{c_2 - c_1}$. We attempt to properly set the value of $\lambda$ such that the $\max$ term equals zero. We find that if we set $\frac{\lambda_1}{\lambda_2} = \frac{\mu_2 - \mu_1}{c_2 - c_1}$,

$$\mathrm{KL}_1(P^{\boldsymbol{\pi}}_{\nu_1,T} \| P^{\boldsymbol{\pi}}_{\nu_2,T})T - \lambda_1 c_1 T - \lambda_2 \left( \sum_{a \in [K]} \mu_a \pi^*_a - \mu_1 \right) T$$

$$= -\lambda_1 c_1 T - \lambda_2 \left( \sum_{a \in [K]} \mu_a \pi^*_a - \mu_1 \right) T$$

$$= -\lambda_1 c_1 T - \lambda_2 (\mu_1 - \mu_2) \frac{c_1}{c_2 - c_1} T = 0$$

and

$$\mathrm{KL}_2(P^{\boldsymbol{\pi}}_{\nu_1,T} \| P^{\boldsymbol{\pi}}_{\nu_2,T})T - \lambda_1 c_2 T - \lambda_2 \left( \sum_{a \in [K]} \mu_a \pi^*_a - \mu_2 \right) T$$

$$= -\lambda_1 c_2 T - \lambda_2 \left( \sum_{a \in [K]} \mu_a \pi^*_a - \mu_2 \right) T$$

$$= -\lambda_1 c_2 T - \lambda_2 (\mu_1 - \mu_2) \frac{c_2}{c_2 - c_1} T = 0.$$

Thus, if we set $\lambda_1 = \lambda_1'$ and $\lambda_2 = \lambda_2'$, where

$$\lambda_1' = \frac{\mu_2 - \mu_1}{c_2 - c_1}\lambda_2', \quad \lambda_2' = \frac{\text{KL}_3(P^{\boldsymbol{\pi}}_{\nu_1,T} || P^{\boldsymbol{\pi}}_{\nu_2,T})}{c_3\frac{\mu_2-\mu_1}{c_2-c_1} + \sum_{a\in[K]}\mu_a\pi_a^* - \mu_3},$$

the optimal value of the dual program

$$\min_{\boldsymbol{\lambda}\succeq 0} \quad \lambda_1\delta_T + \lambda_2 CT^\beta + \max_{a'\in[K]}\left\{\text{KL}_{a'}(P^{\boldsymbol{\pi}}_{\nu_1,T} || P^{\boldsymbol{\pi}}_{\nu_2,T})T - \lambda_1 c_{a'}T - \lambda_2\left(\sum_{a\in[K]}\mu_a\pi_a^* - \mu_{a'}\right)T\right\}$$

$$\leq \lambda_1'\delta_T + \lambda_2' CT^\beta + \max_{a'\in[K]}\left\{\text{KL}_{a'}(P^{\boldsymbol{\pi}}_{\nu_1,T} || P^{\boldsymbol{\pi}}_{\nu_2,T})T - \lambda_1' c_{a'}T - \lambda_2'\left(\sum_{a\in[K]}\mu_a\pi_a^* - \mu_{a'}\right)T\right\}$$

$$= \lambda_1'\delta_T + \lambda_2' CT^\beta$$

$$= \frac{\Delta^2}{20\sigma'^2}\left(\frac{\delta_T}{10} + CT^\beta\right).$$

So we can conclude that

$$\sup_{\boldsymbol{\pi}\in\Pi^{\text{any}}(\beta,\boldsymbol{\delta})}\text{KL}(\mathbb{P}^{\boldsymbol{\pi}}_{\nu_1} || \mathbb{P}^{\boldsymbol{\pi}}_{\nu_2}) \leq \frac{\Delta^2}{20\sigma'^2}\left(\frac{\delta_T}{10} + CT^\beta\right).$$

As a result, the minimax error lower bound is derived as follows:

$$\inf_{\substack{\boldsymbol{\pi}\in\Pi^{\text{any}}(\beta,\boldsymbol{\delta})\\ \hat{\theta}\in\widehat{\Theta}}}\sup_{\nu\in\mathcal{V}}\mathbb{E}^{\boldsymbol{\pi}}_{\nu}[||\hat{\theta} - \theta_\nu^*||_2]$$

$$= \inf_{\substack{\boldsymbol{\pi}\in\Pi^{\text{any}}(\beta,\boldsymbol{\delta})\\ \hat{\theta}\in\widehat{\Theta}}}\sup_{\nu\in\mathcal{V}}\mathbb{E}^{\boldsymbol{\pi}}_{\nu}[||\hat{\theta} - \theta_\nu^*||_2 \mid ||\hat{\theta} - \theta_\nu^*||_2 \geq \Delta]\mathbb{P}^{\boldsymbol{\pi}}_{\nu}(||\hat{\theta} - \theta_\nu^*||_2 \geq \Delta)$$

$$\qquad + \mathbb{E}^{\boldsymbol{\pi}}_{\nu}[||\hat{\theta} - \theta_\nu^*||_2 \mid ||\hat{\theta} - \theta_\nu^*||_2 < \Delta]\mathbb{P}^{\boldsymbol{\pi}}_{\nu}(||\hat{\theta} - \theta_\nu^*||_2 < \Delta)$$

$$\geq \inf_{\substack{\boldsymbol{\pi}\in\Pi^{\text{any}}(\beta,\boldsymbol{\delta})\\ \hat{\theta}\in\widehat{\Theta}}}\sup_{\nu\in\mathcal{V}}\mathbb{E}^{\boldsymbol{\pi}}_{\nu}[||\hat{\theta} - \theta_\nu^*||_2 \mid ||\hat{\theta} - \theta_\nu^*||_2 \geq \Delta]\mathbb{P}^{\boldsymbol{\pi}}_{\nu}(||\hat{\theta} - \theta_\nu^*||_2 \geq \Delta)$$

$$\geq \Delta\inf_{\substack{\boldsymbol{\pi}\in\Pi^{\text{any}}(\beta,\boldsymbol{\delta})\\ \hat{\theta}\in\widehat{\Theta}}}\sup_{\nu\in\mathcal{V}}\mathbb{P}^{\boldsymbol{\pi}}_{\nu}(||\hat{\theta} - \theta_\nu^*||_2 \geq \Delta)$$

$$\geq \frac{\Delta}{2}\left[1 - \sqrt{\frac{1}{2}\sup_{\boldsymbol{\pi}\in\Pi^{\text{any}}(\beta,\boldsymbol{\delta})}\text{KL}(\mathbb{P}^{\boldsymbol{\pi}}_{\nu_1} || \mathbb{P}^{\boldsymbol{\pi}}_{\nu_2})}\right].$$

We set $\Delta = \frac{10\sigma'}{\sqrt{\delta_T + 10CT^\beta}}$, and then observe that

$$\inf_{\substack{\boldsymbol{\pi}\in\Pi^{\text{any}}(\beta,\boldsymbol{\delta})\\ \hat{\theta}\in\widehat{\Theta}}}\sup_{\nu\in\mathcal{V}}\mathbb{E}^{\boldsymbol{\pi}}_{\nu}[||\hat{\theta} - \theta_\nu^*||_2] \geq \frac{\Delta}{2}\left[1 - \sqrt{\frac{1}{2}\cdot\frac{1}{2}}\right] = \frac{\Delta}{4} = \frac{5\sigma'}{2\sqrt{\delta_T + 10CT^\beta}}.$$

## C. Theoretical Analysis for Phase I

### C.1. Statistical Estimation Error Upper Bound

In this part, we aim to derive an upper bound for the statistical estimation error of our proposed SERMiSC algorithm. In order to maximize the utility of each arm in our arm set for obtaining accurate estimations of the true parameter $\theta^*$ across all dimensions, the algorithm must ensure that every arm is sampled a sufficient number of times. Specifically, the expected number of pulls for each arm should be proportional to the length of Phase I, $T_1$.

Recall that $N_i(t) = \sum_{s=1}^t \mathbb{I}\{a_s = i\}$. Given the policy $\boldsymbol{\pi}$ in our algorithm and the fixed sequence $\{q_s\}_{s\in[T_1]}$, it can be

shown that the expected number of pulls for any arm $i \in [K]$ satisfies

$$
\begin{aligned}
\mathbb{E}[N_i(T_1)] =& \mathbb{E}\Big[ \sum_{t=1}^{T_1} \mathbb{I}\{a_t = i\} \Big] = \sum_{t=1}^{T_1} \mathbb{E}\Big[ \mathbb{E}\Big[ \mathbb{I}\{a_t = i\} \Big| \mathcal{H}_{t-1} \Big] \Big] \\
=& \sum_{t=1}^{T_1} \mathbb{E}[\pi_t(i)] \\
\geq& \sum_{t=1}^{T_1} \mathbb{E}[q_t \pi'(i)] = \pi'(i) \sum_{t=1}^{T_1} q_t.
\end{aligned}
$$

The following lemma characterizes the expected estimation error in terms of a fixed vector of sample sizes $\mathbf{N}(T_1)$.

**Lemma C.1.** *Let $\mathcal{X} = \{x_1, ..., x_K\} \subset \mathbb{R}^d$ be a set of $K$ feature vectors. Consider a linear regression model where each feature vector $x_i$ is sampled $N_i$ times, resulting in a total of $N = \sum_{i=1}^{K} N_i$ observations. The observed response for the $t$-th sample associated with feature $x_{a_t}$ is given by:*

$$
y_t = \langle x_{a_t}, \theta^* \rangle + \eta_t, \quad t = 1, ..., N
$$

*where $\theta^*$ is the unknown parameter, and $\eta_t$ are zero-mean homoscedastic conditionally $\sigma$-subGaussian noise terms.*

*Assume that the design matrix is full rank such that the matrix $\sum_{i=1}^{K} N_i x_i x_i^\top$ is invertible. Let $\hat{\theta}$ be the Ordinary Least Squares (OLS) estimator. Then, the expected $\mathcal{L}_2$-norm error of $\hat{\theta}$ is upper bounded such that*

$$
\mathbb{E}_{\mathbf{N}}[||\hat{\theta} - \theta^*||_2^2] \leq \sigma^2 tr\left( \Big( \sum_{i=1}^{K} N_i x_i x_i^\top \Big)^{-1} \right).
$$

*Proof of Lemma C.1.* We define the design matrix $\mathbb{X}^{N \times d}$ such that each row of it corresponds to the feature vector of one observation. Similarly, we define the observed label vector $Y \in \mathbb{R}^N$ and the noise vector $\eta \in \mathbb{R}^N$.

The linear model can be rewritten in its matrix form:

$$
Y = \mathbb{X}\theta^* + \eta.
$$

The OLS estimator is defined to be

$$
\begin{aligned}
\hat{\theta} =& (\mathbb{X}^\top \mathbb{X})^{-1} \mathbb{X}^\top Y \\
=& (\mathbb{X}^\top \mathbb{X})^{-1} \mathbb{X}^\top (\mathbb{X}\theta^* + \eta) \\
=& (\mathbb{X}^\top \mathbb{X})^{-1} (\mathbb{X}^\top \mathbb{X})\theta^* + (\mathbb{X}^\top \mathbb{X})^{-1} \mathbb{X}^\top \eta \\
=& \theta^* + (\mathbb{X}^\top \mathbb{X})^{-1} \mathbb{X}^\top \eta.
\end{aligned}
$$

Obviously, $\mathbb{X}^\top \mathbb{X} = \sum_{t=1}^{N} x_t x_t^\top = \sum_{i=1}^{K} N_i x_i x_i^\top$. Thus, the expected error can be rewritten as

$$
\begin{aligned}
\mathbb{E}_{\mathbf{N}}[||\hat{\theta} - \theta^*||_2^2] =& \mathbb{E}_{\mathbf{N}}\Big[ (\hat{\theta} - \theta^*)^\top (\hat{\theta} - \theta^*) \Big] \\
=& \mathbb{E}_{\mathbf{N}}\Big[ tr\Big( (\hat{\theta} - \theta^*)(\hat{\theta} - \theta^*)^\top \Big) \Big] \\
=& tr\Big( \mathbb{E}_{\mathbf{N}}\Big[ (\hat{\theta} - \theta^*)(\hat{\theta} - \theta^*)^\top \Big] \Big).
\end{aligned}
$$

The covariance matrix satisfies:

$$\mathbb{E}_{N}\left[(\hat{\theta} - \theta^*)(\hat{\theta} - \theta^*)^\top\right]$$
$$=\mathbb{E}_{N}\left[(\mathbb{X}^\top\mathbb{X})^{-1}\mathbb{X}^\top\eta\left((\mathbb{X}^\top\mathbb{X})^{-1}\mathbb{X}^\top\eta\right)^\top\right]$$
$$=\mathbb{E}_{N}\left[(\mathbb{X}^\top\mathbb{X})^{-1}\mathbb{X}^\top\eta\eta^\top\mathbb{X}(\mathbb{X}^\top\mathbb{X})^{-1}\right]$$
$$=(\mathbb{X}^\top\mathbb{X})^{-1}\mathbb{X}^\top\mathbb{E}_{N}[\eta\eta^\top]\mathbb{X}(\mathbb{X}^\top\mathbb{X})^{-1}$$
$$\preceq(\mathbb{X}^\top\mathbb{X})^{-1}\mathbb{X}^\top\sigma^2 I_N\mathbb{X}(\mathbb{X}^\top\mathbb{X})^{-1}$$
$$=\sigma^2(\mathbb{X}^\top\mathbb{X})^{-1}$$

since $\eta_t$ are zero-mean homoscedastic conditionally $\sigma$-subGaussian noise terms. As a consequence, the expected error

$$\mathbb{E}_{N}[||\hat{\theta} - \theta^*||_2^2] =\text{tr}\left(\mathbb{E}_{N}\left[(\hat{\theta} - \theta^*)(\hat{\theta} - \theta^*)^\top\right]\right)$$
$$\leq\text{tr}\left(\sigma^2(\mathbb{X}^\top\mathbb{X})^{-1}\right)$$
$$=\sigma^2\text{tr}\left(\left(\sum_{i=1}^{K} N_i x_i x_i^\top\right)^{-1}\right).$$

$\square$

Knowing the expectation of sample sizes is, unfortunately, insufficient for controlling the expected error $\mathbb{E}[||\hat{\theta} - \theta^*||_2]$. This is because the error is convex with respect to the vector of sample sizes $N(T_1)$, and we cannot use Jensen's inequality to directly compute an **upper bound** for the error. Before proceeding, we recall our assumption that the true parameter $\theta^*$ lies in a reasonably bounded subset $\mathcal{B}_\theta \subset \mathbb{R}^d$ and project our OLS estimator $\hat{\theta}$ onto this feasible set. This is to ensure that the error $||\hat{\theta} - \theta^*||_2$ is uniformly upper bounded. Let a constant $G := \max_{\theta' \in \mathcal{B}_\theta} ||\theta' - \theta^*||_2^2$. Besides, let us define the good event $\mathcal{E}_\epsilon$ when the sample sizes for all arms are large enough:

$$\mathcal{E}_\epsilon := \left\{ N_i(T_1) > (1 - \epsilon)\sum_{t=1}^{T_1} q_t\pi'(i), \quad \forall i \in [K] \right\}.$$

Under this good event $\mathcal{E}_\epsilon$, we can adopt Lemma C.1 to derive an upper bound for the conditional error. We can simultaneously show that the good event indeed happens with a high probability, thus obtaining an ultimate upper bound for the expected error:

$$\text{Pr}(\mathcal{E}_\epsilon) = \text{Pr}\left(\forall i \in [K], N_i(T_1) > (1 - \epsilon)\sum_{t=1}^{T_1} q_t\pi'(i)\right)$$
$$=1 - \text{Pr}\left(\exists i \in [K], N_i(T_1) \leq (1 - \epsilon)\sum_{t=1}^{T_1} q_t\pi'(i)\right)$$
$$\geq1 - \sum_{i=1}^{K}\text{Pr}\left(N_i(T_1) \leq (1 - \epsilon)\sum_{t=1}^{T_1} q_t\pi'(i)\right)$$
$$\geq1 - \sum_{i=1}^{K}\text{Pr}\left(e^{-\chi N_i(T_1)} \geq e^{-\chi(1-\epsilon)\sum_{t=1}^{T_1} q_t\pi'(i)}\right),$$

where $\chi$ is a fixed positive constant. By Markov's inequality, we further have that

$$\text{Pr}(\mathcal{E}_\epsilon) \geq 1 - \sum_{i=1}^{K}\frac{\mathbb{E}[e^{-\chi N_i(T_1)}]}{e^{-\chi(1-\epsilon)\sum_{t=1}^{T_1} q_t\pi'(i)}}.$$

Thus, it suffices to derive an upper bound for $\mathbb{E}[e^{-\chi N_i(T_1)}]$:

$$
\begin{aligned}
&\mathbb{E}[e^{-\chi N_i(T_1)}] \\
=&\mathbb{E}[\mathbb{E}[e^{-\chi N_i(T_1)}|\mathcal{H}_{T_1-1}]] \\
=&\mathbb{E}\left[e^{-\chi\sum_{s=1}^{T_1-1}\mathbb{I}\{a_s=i\}}\mathbb{E}[e^{-\chi\mathbb{I}\{a_{T_1}=i\}}|\mathcal{H}_{T_1-1}]\right] \\
=&\mathbb{E}\left[e^{-\chi\sum_{s=1}^{T_1-1}\mathbb{I}\{a_s=i\}}\mathbb{E}[\mathbb{I}\{a_{T_1}\neq i\}+e^{-\chi}\mathbb{I}\{a_{T_1}=i\}|\mathcal{H}_{T_1-1}]\right] \\
=&\mathbb{E}\left[e^{-\chi\sum_{s=1}^{T_1-1}\mathbb{I}\{a_s=i\}}\left((1-\pi_{T_1}(i))+e^{-\chi}\pi_{T_1}(i)\right)\right] \\
\leq&\mathbb{E}\left[e^{-\chi\sum_{s=1}^{T_1-1}\mathbb{I}\{a_s=i\}}e^{(e^{-\chi}-1)\pi_{T_1}(i)}\right] \\
\leq&\mathbb{E}\left[e^{-\chi\sum_{s=1}^{T_1-1}\mathbb{I}\{a_s=i\}}\right]e^{(e^{-\chi}-1)q_{T_1}\pi'(i)},
\end{aligned}
$$

where $\mathcal{H}_t := \sigma(a_1, \boldsymbol{Z}_1, ..., a_t, \boldsymbol{Z}_t)$. Applying the above inequality recursively, we have

$$
\mathbb{E}[e^{-\chi N_i(T_1)}] \leq e^{(e^{-\chi}-1)\sum_{t=1}^{T_1} q_t\pi'(i)},
$$

and furthermore

$$
\begin{aligned}
\Pr(\mathcal{E}_\epsilon) \geq& 1 - \sum_{i=1}^{K}\exp\left(\left[(e^{-\chi}-1)+\chi(1-\epsilon)\right]\sum_{t=1}^{T_1}q_t\pi'(i)\right) \\
\geq& 1 - \sum_{i=1}^{K}\exp\left(\left[-\epsilon-(1-\epsilon)\ln(1-\epsilon)\right]\sum_{t=1}^{T_1}q_t\pi'(i)\right).
\end{aligned}
$$

As $T_1$ increases, the probability that event $\mathcal{E}_\epsilon$ fails to hold diminishes exponentially. Consequently, the expected estimation error can be upper bounded as follows:

$$
\begin{aligned}
&\mathbb{E}\left[||\hat{\theta}-\theta^*||_2^2\right] \\
=&\mathbb{E}\left[||\hat{\theta}-\theta^*||_2^2\big|\mathcal{E}_\epsilon\right]\Pr(\mathcal{E}_\epsilon)+\mathbb{E}\left[||\hat{\theta}-\theta^*||_2^2\big|\bar{\mathcal{E}}_\epsilon\right]\Pr(\bar{\mathcal{E}}_\epsilon) \\
=&\mathbb{E}\left[\mathbb{E}\left[||\hat{\theta}-\theta^*||_2^2\big|\boldsymbol{N}(T_1),\mathcal{E}_\epsilon\right]\Big|\mathcal{E}_\epsilon\right]\Pr(\mathcal{E}_\epsilon)+\mathbb{E}\left[||\hat{\theta}-\theta^*||_2^2\big|\bar{\mathcal{E}}_\epsilon\right]\Pr(\bar{\mathcal{E}}_\epsilon) \\
=&\mathbb{E}\left[\mathbb{E}_{\boldsymbol{N}(T_1)}\left[||\hat{\theta}-\theta^*||_2^2\right]\Big|\mathcal{E}_\epsilon\right]\Pr(\mathcal{E}_\epsilon)+\mathbb{E}\left[||\hat{\theta}-\theta^*||_2^2\big|\bar{\mathcal{E}}_\epsilon\right]\Pr(\bar{\mathcal{E}}_\epsilon) \\
=&\mathbb{E}\left[\sigma^2\mathrm{tr}\left(\left(\sum_{i=1}^{K}N_i(T_1)x_ix_i^\top\right)^{-1}\right)\Big|\mathcal{E}_\epsilon\right]\Pr(\mathcal{E}_\epsilon)+\mathbb{E}\left[||\hat{\theta}-\theta^*||_2^2\big|\bar{\mathcal{E}}_\epsilon\right]\Pr(\bar{\mathcal{E}}_\epsilon) \\
\leq&\sigma^2\mathrm{tr}\left(\left(\sum_{i=1}^{K}(1-\epsilon)\pi'(i)\sum_{t=1}^{T_1}q_tx_ix_i^\top\right)^{-1}\right)+G\sum_{i=1}^{K}\exp\left(\left[-\epsilon-(1-\epsilon)\ln(1-\epsilon)\right]\sum_{t=1}^{T_1}q_t\pi'(i)\right) \\
\leq&\sigma^2\mathrm{tr}\left(\left(\sum_{i=1}^{K}(1-\epsilon)\pi'(i)T_1\underline{q}x_ix_i^\top\right)^{-1}\right)+G\sum_{i=1}^{K}\exp\left(\left[-\epsilon-(1-\epsilon)\ln(1-\epsilon)\right]T_1\underline{q}\pi'(i)\right).
\end{aligned}
$$

We can select $\epsilon = \frac{1}{10}$ and observe that when $T_1$ is sufficiently large,

$$
\mathbb{E}\left[||\hat{\theta}-\theta^*||_2^2\right] \leq \frac{11\sigma^2}{9\underline{q}T_1}\mathrm{tr}\left(\left(\sum_{i=1}^{K}\pi'(i)x_ix_i^\top\right)^{-1}\right).
$$

Finally, by the definition of variance, we have that

$$
\begin{aligned}
\mathbb{E}\big[||\hat{\theta} - \theta^*||_2\big] &= \sqrt{\mathbb{E}\big[||\hat{\theta} - \theta^*||_2^2\big] - \mathrm{Var}\big[||\hat{\theta} - \theta^*||_2\big]} \\
&\leq \sqrt{\mathbb{E}\big[||\hat{\theta} - \theta^*||_2^2\big]\big]} \\
&\leq \sqrt{\frac{11\sigma^2}{9\underline{q}T_1}\mathrm{tr}\bigg(\Big(\sum_{i=1}^{K}\pi'(i)x_i x_i^\top\Big)^{-1}\bigg)} \\
&= O(T_1^{-\frac{1}{2}}).
\end{aligned}
$$

## C.2. Cost Analysis Results

In this part, we analyze the safety constraint violations of the first phase of the SERMiSC algorithm. As for the cumulative cost, we have that at any time $\tau$,

$$
\begin{aligned}
\mathbb{E}\Big[\sum_{t=1}^{\tau}\sum_{i=1}^{K}Z_t(i)\pi_t(i)\Big] &= \sum_{t=1}^{\tau}\sum_{i=1}^{K}\mathbb{E}\Big[Z_t(i)\pi_t(i)\Big] \\
&= \sum_{t=1}^{\tau}\sum_{i=1}^{K}\mathbb{E}\Big[c_i\pi_t(i)\Big] = \mathbb{E}\Big[\sum_{i=1}^{K}c_i\sum_{t=1}^{\tau}\pi_t(i)\Big] \\
&= \sum_{i=1}^{K}c_i\mathbb{E}\Big[\sum_{t=1}^{\tau}\Big[q_t\pi'(i) + (1-q_t)\frac{e^{-\zeta_t\hat{C}_{t-1}(i)}}{\sum_{j\in[K]}e^{-\zeta_t\hat{C}_{t-1}(j)}}\Big]\Big],
\end{aligned}
$$

recalling that

$$
\hat{C}_t(i) = \sum_{s=1}^{t}\frac{Z_s(i)}{\pi_s(i)}\mathbb{I}\{a_s = i\}.
$$

By inserting an intermediate term $\sum_{i=1}^{K}c_i\mathbb{E}\Big[\sum_{t=1}^{\tau}(1-q_t)\mathbb{I}\{i = \arg\min_j c_j\}\Big]$, we obtain that

$$
\begin{aligned}
&\mathbb{E}\Big[\sum_{t=1}^{\tau}\sum_{i=1}^{K}Z_t(i)\pi_t(i)\Big] \\
&= \sum_{i=1}^{K}c_i\sum_{t=1}^{\tau}q_t\pi'(i) + \sum_{i=1}^{K}c_i\mathbb{E}\Big[\sum_{t=1}^{\tau}(1-q_t)\frac{e^{-\zeta_t\hat{C}_{t-1}(i)}}{\sum_{j\in[K]}e^{-\zeta_t\hat{C}_{t-1}(j)}}\Big] \\
&= \sum_{i=1}^{K}c_i\pi'(i)\sum_{t=1}^{\tau}q_t + c_{\min}\sum_{t=1}^{\tau}(1-q_t) \\
&\quad + \sum_{i=1}^{K}c_i\sum_{t=1}^{\tau}(1-q_t)\mathbb{E}\Big[\frac{e^{-\zeta_t\hat{C}_{t-1}(i)}}{\sum_{j\in[K]}e^{-\zeta_t\hat{C}_{t-1}(j)}} - \mathbb{I}\{i = \arg\min_j c_j\}\Big] \\
&\leq \Big[\sum_{i=1}^{K}c_i\pi'(i)\sum_{t=1}^{\tau}q_t + c_{\min}\sum_{t=1}^{\tau}(1-q_t)\Big] \\
&\quad + \sum_{i\neq i_{\min}\wedge c_i>0}c_i\sum_{t=1}^{\tau}(1-q_t)\mathbb{E}\Big[1\wedge e^{-\zeta_t\big(\hat{C}_{t-1}(i)-\hat{C}_{t-1}(i_{\min})\big)}\Big],
\end{aligned}
$$

where $i_{\min} := \arg\min_j c_j$ and $c_{\min} := c_{i_{\min}}$. If we select the values of $\{q_t\}_{t\in[T_1]}$ to be sufficiently small, the first term on the right-hand side (RHS) of the last inequality will be below 0. To proceed with the derivation, we need to analyze the value $\hat{C}_{t-1}(i) - \hat{C}_{t-1}(i_{\min})$, which asymptotically approximates $(t-1)(c_i - c_{\min}) = \Theta(t)$. Therefore, we expect the second

term on the RHS of the inequality to decrease at an exponential rate. Besides, we only consider arms whose costs $c_i$ are non-negative.

We restate Bernstein's inequality, which is useful in controlling the values of $\hat{C}_{t-1}(i) - \hat{C}_{t-1}(i_{\min})$ for the analysis of cumulative cost.

**Lemma C.2** (Bernstein's Inequality (Simchi-Levi & Wang, 2025)). *Let $X_1, X_2, \cdots$ be a martingale difference sequence, such that $|X_t| \leq \alpha_t$ for a non-decreasing deterministic sequence $\alpha_1, \alpha_2, \cdots$ with probability 1. Let $M_t := \sum_{s=1}^{t} X_s$ be martingale. Let $\bar{U}_1, \bar{U}_2, \cdots$ be deterministic upper bounds on the variance $U_t := \sum_{s=1}^{t} \mathbb{E}[X_s^2 | X_1, \cdots, X_{s-1}]$ of the martingale $M_t$, such that $\bar{U}_t$-s satisfy $\sqrt{\frac{\ln(\frac{2}{\delta})}{(e-2)\bar{U}_t}} \leq \frac{1}{\alpha_t}$. Then with probability greater than $1 - \delta$ for all $t$:*

$$|M_t| \leq 2\sqrt{(e-2)\bar{U}_t \ln \frac{2}{\delta}}.$$

Define the martingale $M_t(i) = \sum_{s=1}^{t} \frac{Z_s(i)}{\pi_s(i)} \mathbb{I}\{a_s = i\} - c_i t$ for arm $i$ at time $t$. Assume that the cost random variable $Z_t$ is almost surely bounded, i.e., $|Z_t| \leq \bar{Z}$ a.s.. We have that the martingale difference $X_t$ is also almost surely upper bounded:

$$
\begin{aligned}
|X_t(i)| &= \left| \frac{Z_t(i)}{\pi_t(i)} \mathbb{I}\{a_t = i\} - c_i \right| \\
&= \left| \frac{Z_t(i) \mathbb{I}\{a_t = i\} - c_i \pi_t(i) + c_i \mathbb{I}\{a_t = i\} - c_i \mathbb{I}\{a_t = i\}}{\pi_t(i)} \right| \\
&\leq \mathbb{I}\{a_t = i\} \frac{|Z_t(i) - c_i|}{\pi_t(i)} + \frac{|c_i|}{\pi_t(i)} |\pi_t(i) - \mathbb{I}\{a_t = i\}| \\
&\leq \frac{|Z_t(i) - c_i| + |c_i|}{\pi_t(i)} \\
&\leq \frac{\bar{Z} + 2\max_{j \in [K]} |c_j|}{q_t \pi'(i)} \leq \frac{\bar{Z} + 2\bar{c}}{q_t \pi'(i)}.
\end{aligned}
$$

To apply Bernstein's inequality, we still need a valid upper bound on the variance of the martingale $M_t$. For arm $i$ at time $t$, the variance is upper bounded as

$$
\begin{aligned}
U_t(i) &= \sum_{s=1}^{t} \mathbb{E}\left[ \left( \frac{Z_s(i)}{\pi_s(i)} \mathbb{I}\{a_s = i\} - c_i \right)^2 \bigg| X_1(i), ..., X_{s-1}(i) \right] \\
&= \sum_{s=1}^{t} \mathbb{E}\left[ \mathbb{E}\left[ \left( \frac{Z_s(i)}{\pi_s(i)} \mathbb{I}\{a_s = i\} - c_i \right)^2 \bigg| \mathcal{H}_{s-1} \right] \bigg| X_1(i), ..., X_{s-1}(i) \right] \\
&= \sum_{s=1}^{t} \mathbb{E}\left[ \mathbb{E}\left[ \frac{Z_s^2(i)}{\pi_s^2(i)} \mathbb{I}\{a_s = i\} + c_i^2 - 2c_i \frac{Z_s(i)}{\pi_s(i)} \mathbb{I}\{a_s = i\} \bigg| \mathcal{H}_{s-1} \right] \bigg| X_1(i), ..., X_{s-1}(i) \right] \\
&= \sum_{s=1}^{t} \mathbb{E}\left[ \mathbb{E}\left[ \frac{Z_s^2(i)}{\pi_s^2(i)} \mathbb{I}\{a_s = i\} \bigg| \mathcal{H}_{s-1} \right] - c_i^2 \bigg| X_1(i), ..., X_{s-1}(i) \right] \\
&\leq \sum_{s=1}^{t} \mathbb{E}\left[ \frac{\bar{Z}^2}{\pi_s^2(i)} \pi_s(i) - c_i^2 \bigg| X_1(i), ..., X_{s-1}(i) \right] \\
&\leq \sum_{s=1}^{t} \frac{\bar{Z}^2}{q_s \pi'(i)}.
\end{aligned}
$$

Therefore, we set the upper bound $\bar{U}_t(i)$ for the variance $U_t(i)$ to be

$$\bar{U}_t(i) := \max \left\{ \sum_{s=1}^{t} \frac{\bar{Z}^2}{q_s \pi'(i)}, \left( \bar{Z} + 2\bar{c} \right)^2 \frac{\ln \frac{2}{\varpi}}{(e-2) q_t^2 \pi'^2(i)} \right\} \geq U_t(i),$$

where the exact value of the confidence $\varpi \in (0, 1)$ will be specified later. It can be easily verified that this definition of $\bar{U}_t(i)$ satisfies

$$\sqrt{\frac{\ln \frac{2}{\varpi}}{(e-2)\bar{U}_t(i)}} \leq \sqrt{\frac{q_t^2 \pi'^2(i)(e-2)\ln \frac{2}{\varpi}}{(e-2)\left(\bar{Z} + 2\bar{c}\right)^2 \ln \frac{2}{\varpi}}}$$

$$= \frac{q_t \pi'(i)}{\bar{Z} + 2\bar{c}}, \quad \forall t.$$

Thus, by Bernstein's inequality, with a probability of at least $1 - \varpi$ for any $t$:

$$|M_t(i)| \leq 2\sqrt{\max\left\{(e-2)\ln\frac{2}{\varpi}\sum_{s=1}^{t}\frac{\bar{Z}^2}{q_s\pi'(i)}, \left(\bar{Z} + 2\bar{c}\right)^2 \frac{(\ln\frac{2}{\varpi})^2}{q_t^2\pi'^2(i)}\right\}}.$$

To derive an upper bound that is of a clear $O(\sqrt{t})$ order, we notice that $|M_t(i)|$ can be further upper bounded as

$$|M_t(i)|$$

$$\leq 2\sqrt{\max\left\{(e-2)\frac{\bar{Z}^2}{\pi'(i)}\ln\frac{2}{\varpi}, \left(\bar{Z} + 2\bar{c}\right)^2 \frac{(\ln\frac{2}{\varpi})^2}{\pi'^2(i)\sum_{s=1}^{t}\frac{q_t^2}{q_s}}\right\}\sum_{s=1}^{t}\frac{1}{q_s}}$$

$$\leq 2\sqrt{\max\left\{(e-2)\frac{\bar{Z}^2}{\pi'(i)}\ln\frac{2}{\varpi}, \left(\bar{Z} + 2\bar{c}\right)^2 \frac{(\ln\frac{2}{\varpi})^2 q_1}{\pi'^2(i)\underline{q}^2}\right\}\sum_{s=1}^{t}\frac{1}{q_s}}$$

$$= 2\sqrt{\left(\bar{Z} + 2\bar{c}\right)^2 \frac{(\ln\frac{2}{\varpi})^2 q_1}{\pi'^2(i)\underline{q}^2}\sum_{s=1}^{t}\frac{1}{q_s}}$$

$$= \frac{2\left(\bar{Z} + 2\bar{c}\right)\ln\frac{2}{\varpi}}{\pi'(i)\underline{q}}\sqrt{q_1\sum_{s=1}^{t}\frac{1}{q_s}}$$

when $\varpi \leq 2e^{-(e-2)(\frac{\bar{Z}}{\bar{Z}+2\bar{c}})^2\pi'(i)\frac{q^2}{q_1}}$. $\underline{q}$ is a uniform lower bound for the value of $q_t$. As a summary, for a small enough $\varpi$, we have that with a probability of at least $1 - K\varpi$,

$$\left|\hat{C}_t(i) - c_i t\right| \leq \frac{2\left(\bar{Z} + 2\bar{c}\right)\ln\frac{2}{\varpi}}{\pi'(i)\underline{q}}\sqrt{q_1\sum_{s=1}^{t}\frac{1}{q_s}}, \quad \forall i \in [K], t.$$

We denote this event as $\mathcal{E}_\varpi$. When this event happens, the value of $\hat{C}_{t-1}(i) - \hat{C}_{t-1}(i_{\min})$ can be effectively controlled. We

have that

$$\hat{C}_{t-1}(i) - \hat{C}_{t-1}(i_{\min})$$
$$= c_i(t-1) - c_{\min}(t-1)$$
$$\quad + \hat{C}_{t-1}(i) - c_i(t-1) - \hat{C}_{t-1}(i_{\min}) + c_{\min}(t-1)$$
$$\geq c_i(t-1) - c_{\min}(t-1)$$
$$\quad - |\hat{C}_{t-1}(i) - c_i(t-1)| - |\hat{C}_{t-1}(i_{\min}) - c_{\min}(t-1)|$$
$$\geq c_i(t-1) - c_{\min}(t-1)$$
$$\quad - \frac{2\big(\bar{Z} + 2\bar{c}\big)\ln\frac{2}{\varpi}}{\pi'(i)\underline{q}}\sqrt{q_1\sum_{s=1}^{t-1}\frac{1}{q_s}} - |\hat{C}_{t-1}(i_{\min}) - c_{\min}(t-1)|$$
$$\geq c_i(t-1) - c_{\min}(t-1)$$
$$\quad - \frac{2\big(\bar{Z} + 2\bar{c}\big)\ln\frac{2}{\varpi}}{\underline{q}}\Big(\frac{1}{\pi'(i)} + \frac{1}{\pi'(i_{\min})}\Big)\sqrt{q_1\sum_{s=1}^{t-1}\frac{1}{q_s}}.$$

We define

$$B(i) = 2\big(\bar{Z} + 2\bar{c}\big)\Big(\frac{1}{\pi'(i)} + \frac{1}{\pi'(i_{\min})}\Big)\sqrt{q_1}/\underline{q}.$$

Before deriving our final bound for the value of $\mathbb{E}\Big[\sum_{t=1}^{\tau}\sum_{i=1}^{K} Z_t(i)\pi_t(i)\Big]$, we first use the result above to control the following term,

$$\sum_{t=1}^{\tau}(1 - q_t)\mathbb{E}\Big[1 \wedge e^{-\zeta_t\big(\hat{C}_{t-1}(i) - \hat{C}_{t-1}(i_{\min})\big)}\Big]$$
$$= \sum_{t=1}^{\tau}(1 - q_t)\mathbb{E}\Big[1 \wedge e^{-\zeta_t\big(\hat{C}_{t-1}(i) - \hat{C}_{t-1}(i_{\min})\big)}\Big(\mathbb{I}\{\mathcal{E}_{\varpi}\} + \mathbb{I}\{\bar{\mathcal{E}}_{\varpi}\}\Big)\Big]$$
$$\leq \sum_{t=1}^{\tau}(1 - q_t)\Big(\mathbb{E}e^{-\zeta_t\big(\hat{C}_{t-1}(i) - \hat{C}_{t-1}(i_{\min})\big)}\mathbb{I}\{\mathcal{E}_{\varpi}\} + \Pr\{\bar{\mathcal{E}}_{\varpi}\}\Big)$$
$$\leq \sum_{t=1}^{\tau}(1 - q_t)\Big(\mathbb{E}e^{-\zeta_t\big(\hat{C}_{t-1}(i) - \hat{C}_{t-1}(i_{\min})\big)}\mathbb{I}\{\mathcal{E}_{\varpi}\} + K\varpi\Big)$$
$$\leq K\varpi\sum_{t=1}^{\tau}(1 - q_t) + \sum_{t=1}^{\tau}(1 - q_t)e^{-\zeta_t\big((c_i - c_{\min})(t-1) - B(i)\ln\frac{2}{\varpi}\sqrt{\sum_{s=1}^{t-1}\frac{1}{q_s}}\big)}$$
$$= K\varpi\sum_{t=1}^{\tau}(1 - q_t) + \sum_{t=1}^{\tau}(1 - q_t)e^{-\zeta_t(c_i - c_{\min})(t-1)} \cdot \Big(\frac{2}{\varpi}\Big)^{\zeta_t B(i)\sqrt{\sum_{s=1}^{t-1}\frac{1}{q_s}}}$$
$$\leq K\varpi\sum_{t=1}^{\tau}(1 - q_t) + \max_{\tilde{s}\geq 1}\Big(\frac{2}{\varpi}\Big)^{\zeta_{\tilde{s}} B(i)\sqrt{\sum_{s=1}^{\tilde{s}-1}\frac{1}{q_s}}} \cdot \sum_{t=1}^{\tau}(1 - q_t)e^{-\zeta_t(c_i - c_{\min})(t-1)}.$$

We continue our derivation by recalling our choice of

$$\zeta_1 = D\Big(\max_{j\in[K]} B(j)\Big)^{-1}, \quad \zeta_t = D\Big(\max_{j\in[K]} B(j)\sqrt{\sum_{s=1}^{t-1}\frac{1}{q_s}}\Big)^{-1}, \quad \forall t \geq 2,$$

with a constant $D = 1/\ln T_1 > 0$. It can be easily observed that for any $t \geq 2$,

$$\zeta_t \geq \frac{D}{\max_{j\in[K]} B(j)}\sqrt{\frac{\underline{q}}{t-1}}.$$

Now, with a trivial requirement that $\varpi < 2$, the above term can be further upper bounded as

$$\sum_{t=1}^{\tau}(1-q_t)\mathbb{E}\Big[1 \wedge e^{-\zeta_t\big(\hat{C}_{t-1}(i)-\hat{C}_{t-1}(i_{\min})\big)}\Big]$$

$$\leq K\varpi\sum_{t=1}^{\tau}(1-q_t) + \Big(\frac{2}{\varpi}\Big)^D \cdot \sum_{t=1}^{\tau}(1-q_t)e^{-\zeta_t(c_i-c_{\min})(t-1)}$$

$$\leq K\varpi\sum_{t=1}^{\tau}(1-q_t) + \Big(\frac{2}{\varpi}\Big)^D \cdot \sum_{t=1}^{\tau}(1-q_t)e^{-\frac{(c_i-c_{\min})D\sqrt{\underline{q}}}{\max_{j\in[K]}B(j)}(t-1)^{\frac{1}{2}}}$$

$$\leq K\varpi\sum_{t=1}^{\tau}(1-q_t) + \Big(\frac{2}{\varpi}\Big)^D (1-\underline{q})\sum_{t=1}^{\tau}e^{-F(i)(t-1)^{\frac{1}{2}}}$$

$$\leq K\varpi\sum_{t=1}^{\tau}(1-q_t) + \Big(\frac{2}{\varpi}\Big)^D (1-\underline{q})\Big[1+\sum_{t=1}^{\tau}e^{-F(i)t^{\frac{1}{2}}}\Big]$$

$$\leq K\varpi\sum_{t=1}^{\tau}(1-q_t) + \Big(\frac{2}{\varpi}\Big)^D (1-\underline{q})\Big[1+\int_0^{\tau}e^{-F(i)x^{\frac{1}{2}}}dx\Big]$$

$$= K\varpi\sum_{t=1}^{\tau}(1-q_t) + \Big(\frac{2}{\varpi}\Big)^D (1-\underline{q})\Big[1+\frac{2}{F(i)^2}\int_0^{F(i)\tau^{\frac{1}{2}}}e^{-u}u\,du\Big]$$

$$\leq K\varpi\sum_{t=1}^{\tau}(1-q_t) + \Big(\frac{2}{\varpi}\Big)^D (1-\underline{q})\Big[1+\frac{2}{F(i)^2}\Gamma(2)\Big]$$

with $F(i) := \frac{(c_i-c_{\min})D\sqrt{\underline{q}}}{\max_{j\in[K]}B(j)}$. $\Gamma(z) = \int_0^{\infty}t^{z-1}e^{-t}dt$ is the Gamma function. $\Gamma(2) = (2-1)! = 1$. By choosing

$$\varpi = \Big[\frac{2^D D(1-\underline{q})(1+\frac{2}{F(i)^2})}{K\sum_{t=1}^{\tau}(1-q_t)}\Big]^{\frac{1}{1+D}}$$

for arm $i$, we obtain that

$$\sum_{t=1}^{\tau}(1-q_t)\mathbb{E}\Big[1 \wedge e^{-\zeta_t\big(\hat{C}_{t-1}(i)-\hat{C}_{t-1}(i_{\min})\big)}\Big]$$

$$\leq \Big(D^{\frac{1}{1+D}} + D^{-\frac{D}{D+1}}\Big)\Big[2K\sum_{t=1}^{\tau}(1-q_t)\Big]^{\frac{D}{D+1}}\Big[(1-\underline{q})(1+\frac{2}{F(i)^2})\Big]^{\frac{1}{D+1}},$$

which enables us to derive an upper bound for $\mathbb{E}\Big[\sum_{t=1}^{\tau}\sum_{i=1}^{K}Z_t(i)\pi_t(i)\Big]$:

$$\mathbb{E}\Big[\sum_{t=1}^{\tau}\sum_{i=1}^{K}Z_t(i)\pi_t(i)\Big]$$

$$\leq \sum_{i=1}^{K}c_i\pi'(i)\sum_{t=1}^{\tau}q_t + c_{\min}\sum_{t=1}^{\tau}(1-q_t)$$

$$+ \sum_{i\neq i_{\min}\wedge c_i>0}c_i\Big(D^{\frac{1}{1+D}} + D^{-\frac{D}{D+1}}\Big)\Big[2K\sum_{t=1}^{\tau}(1-q_t)\Big]^{\frac{D}{D+1}}\Big[(1-\underline{q})(1+\frac{2}{F(i)^2})\Big]^{\frac{1}{D+1}}$$

$$\leq \sum_{i=1}^{K}c_i\pi'(i)\sum_{t=1}^{\tau}q_t + c_{\min}\sum_{t=1}^{\tau}(1-q_t)$$

$$+ 2\sum_{i\neq i_{\min}\wedge c_i>0}c_i\Big[2K\sum_{t=1}^{\tau}(1-q_t)\Big]^{\frac{D}{D+1}}\Big[(1-\underline{q})(1+\frac{2}{F(i)^2})\Big]^{\frac{1}{D+1}}$$

when $T_1$ is sufficiently large. To gain a clearer understanding of how the cumulative cost scales with $T_1$, we set $\tau = T_1$ and have that

$$\mathbb{E}\Big[\sum_{t=1}^{T_1}\sum_{i=1}^{K} Z_t(i)\pi_t(i)\Big]$$

$$\leq \sum_{i=1}^{K} c_i\pi'(i)\sum_{t=1}^{T_1} q_t + c_{\min}\sum_{t=1}^{T_1}(1-q_t)$$

$$+ 2\sum_{i\neq i_{\min}\wedge c_i>0} c_i\Big[2K\sum_{t=1}^{T_1}(1-\underline{q})\Big]^{\frac{D}{D+1}}\Big[(1-\underline{q})(1+\frac{2}{F(i)^2})\Big]^{\frac{1}{D+1}}$$

$$= \sum_{i=1}^{K} c_i\pi'(i)\sum_{t=1}^{T_1} q_t + c_{\min}\sum_{t=1}^{T_1}(1-q_t)$$

$$+ 2\sum_{i\neq i_{\min}\wedge c_i>0} c_i(2KT_1)^{\frac{D}{D+1}}\Big[1+\frac{2}{F(i)^2}\Big]^{\frac{1}{D+1}}(1-\underline{q}).$$

Thus, we have the following asymptotic upper bound

$$\mathbb{E}\Big[\sum_{t=1}^{T_1}\sum_{i=1}^{K} Z_t(i)\pi_t(i)\Big] - \Big[\sum_{i=1}^{K} c_i\pi'(i)\sum_{t=1}^{T_1} q_t + c_{\min}\sum_{t=1}^{T_1}(1-q_t)\Big] = O\big((\ln T_1)^2\big),$$

by noting that $T_1^{\frac{D}{D+1}} = T_1^{\frac{1}{1+\ln T_1}} = e^{\frac{\ln T_1}{1+\ln T_1}} \leq e$.

## D. Theoretical Analysis for Phase II

### D.1. Regret Upper Bound

In this part, we derive an upper bound for the regret of the second phase of our proposed SERMiSC algorithm. To maintain notational simplicity, we reset the time scale for Phase II, beginning the index from $t = 1$. Recall that the algorithm implements the following arm selection strategy in Phase II:

$$a_t = \arg\max_{i\in[K]} \hat{r}_t(i) - \frac{Z_t(i)Q(t)}{V_t}.$$

The arm selection strategy is based on maximizing a modified utility score, balancing optimistic reward estimation with a dynamic safety penalty. The first term,

$$\hat{r}_t(i) = \text{clip}_{[-1,1]}\Big[\max_{\theta\in C_{t-1}(\varrho)}\langle\theta, x_i\rangle\Big],$$

represents the optimistic estimate of the expected reward, typically implemented as an Upper Confidence Bound (UCB) type approach to drive exploration and ensure regret minimization. The second term constitutes a dynamic penalty directly related to the safety constraint. Here, $Z_t(i)$ is the stochastic cost incurred by pulling arm $i$, reflecting the immediate risk of constraint violation. Crucially, $Q(t)$ acts as an adaptively designed Lagrange multiplier, dynamically adjusting the penalty's weight based on the observed cumulative safety performance. Formally,

$$Q(t+1) = \Big[Q(t) + \sum_{i\in[K]} Z_t(i)\mathbb{I}\{a_t = i\} + \epsilon_t\Big]^+.$$

When the system approaches or exceeds the safety boundary, $Q(t)$ increases significantly, making the penalty term dominant. This mechanism ensures that the overall strategy maintains safety-aware prudence, compelling the agent to select safer arms (those with lower $Z_t(i)$) when necessary, thereby upholding the primary safety constraint while striving for optimal rewards.

Define the Lyapunov function with respect to $Q(t)$:

$$L(t) = \frac{1}{2}Q(t)^2.$$

Our primary objective in this part is to derive a sharp upper bound on the expected cumulative regret of the proposed algorithm in Phase II. This derivation employs a technique of policy decomposition to break down the total regret into several manageable terms. The regret can be decomposed as follows:

$$
\begin{aligned}
\mathcal{R}(\tau) =& \tau \sum_{j \in [K]} \pi_j^* \mu_j - \mathbb{E}\left[ \sum_{t=1}^{\tau} \sum_{j \in [K]} \mathbb{I}\{a_t = j\} R_t(j) \right] \\
=& \tau \sum_{j \in [K]} \pi_j^* \mu_j - \mathbb{E}\left[ \sum_{t=1}^{\tau} \sum_{j \in [K]} \mathbb{I}\{a_t = j\} \mu_j \right] \\
=& \sum_{t=1}^{\tau} \sum_{j \in [K]} (\pi_j^* - \pi_{\epsilon_t,\delta}^*(j)) \mu_j + \mathbb{E}\left[ \sum_{t=1}^{\tau} \sum_{j \in [K]} (\pi_{\epsilon_t,\delta}^*(j) - \mathbb{I}\{a_t = j\}) \mu_j \right] \\
=& \sum_{t=1}^{\tau} \sum_{j \in [K]} (\pi_j^* - \pi_{\epsilon_t,\delta}^*(j)) \mu_j + \mathbb{E}\left[ \sum_{t=1}^{\tau} \sum_{j \in [K]} (\pi_{\epsilon_t,\delta}^*(j) - \mathbb{I}\{a_t = j\}) \hat{r}_t(j) \right] \\
& + \mathbb{E}\left[ \sum_{t=1}^{\tau} \sum_{j \in [K]} \pi_{\epsilon_t,\delta}^*(j)(\mu_j - \hat{r}_t(j)) \right] + \mathbb{E}\left[ \sum_{t=1}^{\tau} \sum_{j \in [K]} \mathbb{I}\{a_t = j\}(\hat{r}_t(j) - \mu_j) \right],
\end{aligned}
$$

where $R_t(j)$ is the random variable representing the reward obtained from pulling arm $j$ at time $t$, and $\pi_{\epsilon_t,\delta}^*$ is an intermediate policy introduced to facilitate theoretical analysis. It is defined such that

$$
\begin{aligned}
\pi_{\epsilon_t,\delta}^* \in \arg\max_{\pi} \quad & \sum_{a \in [K]} \pi_a \mu_a, \\
s.t. \quad & \sum_{a \in [K]} \pi_a c_a + \min\{\epsilon_t, \delta\} \leq 0, \\
& \pi \in \Delta_K.
\end{aligned}
$$

The remainder of the proof is structured as a sequential bounding of these four terms:

- $\mathcal{T}_1$: Policy Gap Term

$$\mathcal{T}_1(\tau) := \sum_{t=1}^{\tau} \sum_{j \in [K]} (\pi_j^* - \pi_{\epsilon_t,\delta}^*(j)) \mu_j$$

  This term captures the fundamental difference in expected reward between the true optimal policy $\pi^*$ and our chosen intermediate policy $\pi_{\epsilon_t,\delta}^*$. It measures the loss due to the sub-optimality of the intermediate policy itself.

- $\mathcal{T}_2$: Lyapunov Drift Term

$$\mathcal{T}_2(\tau) := \mathbb{E}\left[ \sum_{t=1}^{\tau} \sum_{j \in [K]} (\pi_{\epsilon_t,\delta}^*(j) - \mathbb{I}\{a_t = j\}) \hat{r}_t(j) \right]$$

  This is the core term that links the algorithm's actual action sequence $\mathbb{I}\{a_t = j\}$ to the desired sequence dictated by the intermediate policy $\pi_{\epsilon_t,\delta}^*$. We will show that this term, often referred to as the Lyapunov drift term, converges or remains small. Specifically, we will prove that the algorithm's arm selection mechanism drives the selection sequence to closely approximate the distribution of the intermediate policy.

- $\mathcal{T}_3, \mathcal{T}_4$: Estimation Error Terms

$$\mathcal{T}_3(\tau) := \mathbb{E}\left[ \sum_{t=1}^{\tau} \sum_{j \in [K]} \pi_{\epsilon_t,\delta}^*(j)(\mu_j - \hat{r}_t(j)) \right]$$

$$\mathcal{T}_4(\tau) := \mathbb{E}\left[\sum_{t=1}^{\tau} \sum_{j\in[K]} \mathbb{I}\{a_t = j\}(\hat{r}_t(j) - \mu_j)\right]$$

These two terms quantify the error introduced by the difference between the true expected rewards $\mu_j$ and the optimistic estimates $\hat{r}_t(j)$ used by the algorithm. By establishing high-probability confidence bounds on our optimistic estimator $\hat{r}_t(j)$, we will effectively control the magnitude of these terms.

Consider $\mathcal{T}_1(\tau)$.

Recall that strategy $\boldsymbol{\pi}_\delta$ was defined in Assumption 4.6. In order to compute a lower bound for $\sum_{j\in[K]} \pi^*_{\epsilon_t,\delta}(j)\mu_j$, we construct

$$\tilde{\boldsymbol{\pi}}_t = \left(1 - \frac{\min\{\epsilon_t, \delta\}}{\delta}\right)\boldsymbol{\pi}^* + \frac{\min\{\epsilon_t, \delta\}}{\delta}\boldsymbol{\pi}_\delta,$$

which can be validated to be a feasible solution to the following optimization problem:

$$\max_{\boldsymbol{\pi}} \quad \sum_{a\in[K]} \pi_a \mu_a,$$

$$s.t. \quad \sum_{a\in[K]} \pi_a c_a + \min\{\epsilon_t, \delta\} \leq 0,$$

$$\boldsymbol{\pi} \in \Delta_K,$$

because

$$\sum_{a\in[K]} \tilde{\pi}_t(a)c_a + \min\{\epsilon_t, \delta\}$$

$$\leq \left(1 - \frac{\min\{\epsilon_t, \delta\}}{\delta}\right)\sum_{a\in[K]} \pi^*_a c_a + \frac{\min\{\epsilon_t, \delta\}}{\delta}\sum_{a\in[K]} \pi_\delta(a)c_a + \min\{\epsilon_t, \delta\}$$

$$\leq \frac{\min\{\epsilon_t, \delta\}}{\delta}\sum_{a\in[K]} \pi_\delta(a)c_a + \min\{\epsilon_t, \delta\}$$

$$\leq \frac{\min\{\epsilon_t, \delta\}}{\delta}(-\delta) + \min\{\epsilon_t, \delta\}$$

$$=0$$

and

$$\sum_{j\in[K]} \tilde{\pi}_t(j)$$

$$= \sum_{j\in[K]} \left(1 - \frac{\min\{\epsilon_t, \delta\}}{\delta}\right)\pi^*_j + \frac{\min\{\epsilon_t, \delta\}}{\delta}\pi_\delta(j)$$

$$= \left(1 - \frac{\min\{\epsilon_t, \delta\}}{\delta}\right) + \frac{\min\{\epsilon_t, \delta\}}{\delta}$$

$$=1.$$

Thus, $\sum_{j\in[K]} \pi^*_{\epsilon_t,\delta}(j)\mu_j$ is lower bounded such that

$$\sum_{j\in[K]} \pi^*_{\epsilon_t,\delta}(j)\mu_j \geq \sum_{j\in[K]} \tilde{\pi}_t(j)\mu_j.$$

So we have

$$
\sum_{t=1}^{\tau} \sum_{j \in [K]} (\pi_j^* - \pi_{\epsilon_t, \delta}^*(j)) \mu_j
$$

$$
\leq \sum_{t=1}^{\tau} \sum_{j \in [K]} \left[ \pi_j^* - \left( 1 - \frac{\min\{\epsilon_t, \delta\}}{\delta} \right) \pi_j^* - \frac{\min\{\epsilon_t, \delta\}}{\delta} \pi_\delta(j) \right] \mu_j
$$

$$
\leq \sum_{t=1}^{\tau} \sum_{j \in [K]} \left[ \pi_j^* - \left( 1 - \frac{\min\{\epsilon_t, \delta\}}{\delta} \right) \pi_j^* \right] \mu_j
$$

$$
\leq \sum_{t=1}^{\tau} \sum_{j \in [K]} \frac{\min\{\epsilon_t, \delta\}}{\delta} \pi_j^* \mu_j
$$

$$
\leq \sum_{t=1}^{\tau} \frac{\min\{\epsilon_t, \delta\}}{\delta} M,
$$

where $M$ is a uniform upper bound for $\mu_j$: $|\mu_j| \leq M, \forall j \in [K]$.

Consider $\mathcal{T}_2(\tau)$.

We note that the Lyapunov drift yields

$$
L(t+1) - L(t)
$$

$$
\leq \frac{1}{2} \Big[ Q(t) + \sum_{j \in [K]} Z_t(j) \mathbb{I}\{a_t = j\} + \epsilon_t \Big]^2 - \frac{1}{2} Q(t)^2
$$

$$
= Q(t) \Big[ \sum_{j \in [K]} Z_t(j) \mathbb{I}\{a_t = j\} + \epsilon_t \Big] + \frac{1}{2} \Big[ \sum_{j \in [K]} Z_t(j) \mathbb{I}\{a_t = j\} + \epsilon_t \Big]^2
$$

$$
\leq Q(t) \Big[ \sum_{j \in [K]} Z_t(j) \mathbb{I}\{a_t = j\} + \epsilon_t \Big] + (\bar{Z}^2 + \epsilon_t^2)
$$

$$
= Q(t) \Big[ \sum_{j \in [K]} Z_t(j) \mathbb{I}\{a_t = j\} + \epsilon_t \Big] - V_t \sum_{j \in [K]} \hat{r}_t(j) \mathbb{I}\{a_t = j\}
$$

$$
+ V_t \sum_{j \in [K]} \hat{r}_t(j) \mathbb{I}\{a_t = j\} + (\bar{Z}^2 + \epsilon_t^2).
$$

Recall that our arm selection policy $a_t = \arg\max_{i \in [K]} \hat{r}_t(i) - \frac{Z_t(i)Q(t)}{V_t}$ implies that

$$
V_t \hat{r}_{a_t} - Q(t) Z_t(a_t)
$$

$$
\geq V_t \sum_{j \in [K]} \pi_j \hat{r}_t(j) - Q(t) \sum_{j \in [K]} \pi_j Z_t(j), \quad \forall \pi.
$$

and furthermore

$$
Q(t) \sum_{j \in [K]} Z_t(j) \mathbb{I}\{a_t = j\} \leq V_t \hat{r}_{a_t} - V_t \sum_{j \in [K]} \pi_j \hat{r}_t(j) + Q(t) \sum_{j \in [K]} \pi_j Z_t(j).
$$

As a consequence, we are able to link the interested difference term $\pi_j - \mathbb{I}\{a_t = j\}$ to the Lyapunov drift, which is easy to

cancel out when summing over the time horizon. We proceed our derivation for the drift

$$
\begin{aligned}
&L(t+1) - L(t) \\
&\leq Q(t)\Big[ \sum_{j\in[K]} Z_t(j)\mathbb{I}\{a_t = j\} + \epsilon_t\Big] - V_t \sum_{j\in[K]} \hat{r}_t(j)\mathbb{I}\{a_t = j\} \\
&\quad + V_t \sum_{j\in[K]} \hat{r}_t(j)\mathbb{I}\{a_t = j\} + (\bar{Z}^2 + \epsilon_t^2) \\
&\leq Q(t)\Big[ \sum_{j\in[K]} Z_t(j)\pi_j + \epsilon_t\Big] - V_t \sum_{j\in[K]} \pi_j \hat{r}_t(j) \\
&\quad + V_t \sum_{j\in[K]} \hat{r}_t(j)\mathbb{I}\{a_t = j\} + (\bar{Z}^2 + \epsilon_t^2).
\end{aligned}
$$

We take the conditional expectation on both sides and obtain for any arm selection policy $\boldsymbol{\pi}$ at time $t$,

$$
\begin{aligned}
&\mathbb{E}\Big[ \sum_{j\in[K]} (\pi_j - \mathbb{I}\{a_t = j\})\hat{r}_t(j)\Big|\mathcal{H}_t\Big] \\
&\leq \mathbb{E}\Big[ -\frac{L(t+1) - L(t)}{V_t}\Big|\mathcal{H}_t\Big] + \frac{Q(t)}{V_t}\mathbb{E}\Big[ \sum_{j\in[K]} Z_t(j)\pi_j + \epsilon_t\Big|\mathcal{H}_t\Big] + \frac{\bar{Z}^2 + \epsilon_t^2}{V_t} \\
&\leq \mathbb{E}\Big[ \frac{L(t)}{V_t} - \frac{L(t+1)}{V_{t+1}}\Big|\mathcal{H}_t\Big] + \frac{Q(t)}{V_t}\mathbb{E}\Big[ \sum_{j\in[K]} Z_t(j)\pi_j + \epsilon_t\Big|\mathcal{H}_t\Big] + \frac{\bar{Z}^2 + \epsilon_t^2}{V_t} \\
&= \mathbb{E}\Big[ \frac{L(t)}{V_t} - \frac{L(t+1)}{V_{t+1}}\Big|\mathcal{H}_t\Big] + \frac{Q(t)}{V_t}\mathbb{E}\Big[ \sum_{j\in[K]} Z_t(j)\pi_j \\
&\quad + \epsilon_t\Big(\mathbb{I}\{\epsilon_t \leq \delta\} + \mathbb{I}\{\epsilon_t > \delta\}\Big)\Big|\mathcal{H}_t\Big] + \frac{\bar{Z}^2 + \epsilon_t^2}{V_t} \\
&= \mathbb{E}\Big[ \frac{L(t)}{V_t} - \frac{L(t+1)}{V_{t+1}}\Big|\mathcal{H}_t\Big] + \frac{Q(t)}{V_t}\mathbb{E}\Big[ \sum_{j\in[K]} Z_t(j)\pi_j + \epsilon_t\Big|\mathcal{H}_t\Big]\mathbb{I}\{\epsilon_t \leq \delta\} \\
&\quad + \frac{Q(t)}{V_t}\mathbb{E}\Big[ \sum_{j\in[K]} Z_t(j)\pi_j + \epsilon_t\Big|\mathcal{H}_t\Big]\mathbb{I}\{\epsilon_t > \delta\} + \frac{\bar{Z}^2 + \epsilon_t^2}{V_t} \\
&= \mathbb{E}\Big[ \frac{L(t)}{V_t} - \frac{L(t+1)}{V_{t+1}}\Big|\mathcal{H}_t\Big] + \frac{Q(t)}{V_t}\mathbb{E}\Big[ \sum_{j\in[K]} Z_t(j)\pi_j + \epsilon_t\Big|\mathcal{H}_t\Big]\mathbb{I}\{\epsilon_t \leq \delta\} \\
&\quad + \frac{Q(t)}{V_t}\mathbb{E}\Big[ \sum_{j\in[K]} Z_t(j)\pi_j + \delta\Big|\mathcal{H}_t\Big]\mathbb{I}\{\epsilon_t > \delta\} + \frac{\bar{Z}^2 + \epsilon_t^2}{V_t} + \frac{Q(t)}{V_t}(\epsilon_t - \delta)\mathbb{I}\{\epsilon_t > \delta\}.
\end{aligned}
$$

Recall that the benchmark strategy $\boldsymbol{\pi}^*$ is given by

$$
\begin{aligned}
\boldsymbol{\pi}^* \in \arg\max_{\boldsymbol{\pi}} \quad & \sum_{a\in[K]} \pi_a \mu_a, \\
s.t. \quad & \sum_{a\in[K]} \pi_a c_a \leq 0, \\
& \boldsymbol{\pi} \in \Delta_K.
\end{aligned}
$$

Unfortunately, we are unable to directly control the value of $\mathbb{E}\Big[ \sum_{j\in[K]} (\pi_j^* - \mathbb{I}\{a_t = j\})\hat{r}_t(j)|\mathcal{H}_t\Big]$ since $\epsilon_t$ and $\delta$ in the

RHS of the last equality cannot be eliminated. This necessitates the selection of an intermediate strategy $\boldsymbol{\pi}^*_{\epsilon_t, \delta}$ such that

$$\boldsymbol{\pi}^*_{\epsilon_t, \delta} \in \arg\max_{\boldsymbol{\pi}} \quad \sum_{a \in [K]} \pi_a \mu_a,$$

$$s.t. \quad \sum_{a \in [K]} \pi_a c_a + \min\{\epsilon_t, \delta\} \leq 0,$$

$$\boldsymbol{\pi} \in \Delta_K,$$

and then control the difference between $\boldsymbol{\pi}^*$ and $\boldsymbol{\pi}^*_{\epsilon_t, \delta}$. For the chosen strategy $\boldsymbol{\pi} = \boldsymbol{\pi}^*_{\epsilon_t, \delta}$, one can directly verify that both $\mathbb{E}\Big[\sum_{j \in [K]} Z_t(j)\pi_j + \epsilon_t \Big| \mathcal{H}_t\Big]\mathbb{I}\{\epsilon_t \leq \delta\}$ and $\mathbb{E}\Big[\sum_{j \in [K]} Z_t(j)\pi_j + \delta \Big| \mathcal{H}_t\Big]\mathbb{I}\{\epsilon_t > \delta\}$ are smaller than 0.

Now we can calculate the cumulative term of interest at time $\tau$:

$$\mathbb{E}\Big[\sum_{t=1}^{\tau} \sum_{j \in [K]} (\pi^*_{\epsilon_t, \delta}(j) - \mathbb{I}\{a_t = j\})\hat{r}_t(j)\Big]$$

$$\leq \frac{\mathbb{E}[L(1)]}{V_1} - \frac{\mathbb{E}[L(\tau + 1)]}{V_{\tau+1}} + \sum_{t=1}^{\tau} \frac{\bar{Z}^2 + \epsilon_t^2}{V_t} + \sum_{t=1}^{\tau} \frac{\mathbb{E}\big[Q(t)(\epsilon_t - \delta)\mathbb{I}\{\epsilon_t > \delta\}\big]}{V_t}$$

$$\leq \frac{\mathbb{E}[L(1)]}{V_1} - \frac{\mathbb{E}[L(\tau + 1)]}{V_{\tau+1}} + \sum_{t=1}^{\tau} \frac{\bar{Z}^2 + \epsilon_t^2}{V_t} + \sum_{t=1}^{\tau} \frac{t(\bar{Z} + \epsilon_1)(\epsilon_t - \delta)\mathbb{I}\{\epsilon_t > \delta\}}{V_t}$$

$$\leq \sum_{t=1}^{\tau} \frac{\bar{Z}^2 + \epsilon_t^2}{V_t} + \sum_{t=1}^{\tau} \frac{t(\bar{Z} + \epsilon_1)(\epsilon_t - \delta)\mathbb{I}\{\epsilon_t > \delta\}}{V_t},$$

where the second inequality is due to the trivial bound that $Q(t) \leq \sum_{s=1}^{t}(\bar{Z} + \epsilon_s) \leq t(\bar{Z} + \epsilon_1)$, and the last inequality follows from noticing that the constraint violation at the very beginning is surely 0.

Consider $\mathcal{T}_3$ and $\mathcal{T}_4$.

These values can be bounded if $\hat{r}_t(j)$ is close to $\mu_j$, which is characterized by the event $\mathcal{E}_C = \Big\{\forall t \geq 0 : \theta^* \in C_t(\varrho)\Big\}$, where the confidence set

$$C_t(\varrho) := \left\{\theta \in \mathbb{R}^d : ||\hat{\theta}_t - \theta||_{\Sigma_t} \leq \sigma\sqrt{d\log\left(\frac{1 + tL^2/\lambda}{\varrho}\right)} + \sqrt{\lambda}S\right\}$$

where we define

$$\Sigma_t = \lambda I_d + \sum_{s=1}^{t} x_{a_t} x_{a_t}^\top,$$

$$\hat{\theta}_t = \Sigma_t^{-1} \sum_{s=1}^{t} y_{a_t} x_{a_t}$$

and assume that $\theta^*, x_j$ are uniformly bounded: $||\theta^*||_2 \leq S$ and $||x_j||_2 \leq L, \forall j \in [K]$. Theorem 2 in (Abbasi-Yadkori et al., 2011) implies that $\Pr(\mathcal{E}_C) \geq 1 - \varrho$.

Under the event $\mathcal{E}_C$, we have

$$
\begin{aligned}
&\hat{r}_t(j) - \mu_j \\
&\leq \max_{\theta \in C_{t-1}(\varrho)} \langle \theta, x_j \rangle - \langle \theta^*, x_j \rangle := \langle \tilde{\theta}_t - \theta^*, x_j \rangle \\
&= \langle \tilde{\theta}_t - \hat{\theta}_{t-1} + \hat{\theta}_{t-1} - \theta^*, x_j \rangle \\
&\leq ||\tilde{\theta}_t - \hat{\theta}_{t-1}||_{\Sigma_{t-1}} ||x_j||_{\Sigma_{t-1}^{-1}} + ||\hat{\theta}_{t-1} - \theta^*||_{\Sigma_{t-1}} ||x_j||_{\Sigma_{t-1}^{-1}} \\
&\leq 2 \left( \sigma \sqrt{d \log \left( \frac{1 + tL^2/\lambda}{\varrho} \right)} + \sqrt{\lambda} S \right) ||x_j||_{\Sigma_{t-1}^{-1}} \\
&\leq 2 \left( \sigma \sqrt{d \log \left( \frac{1 + tL^2/\lambda}{\varrho} \right)} + \sqrt{\lambda} S \right) \min \left\{ 1, ||x_j||_{\Sigma_{t-1}^{-1}} \right\}
\end{aligned}
$$

by noting that our choice of $\tilde{\theta}_t$ is in $C_{t-1}(\varrho)$ and under the event $\mathcal{E}_C$, we also have $\theta^* \in C_{t-1}(\varrho)$. This observation implies the following upper bound when choosing $\varrho = 1/T$:

$$
\begin{aligned}
&\mathbb{E}\left[ \sum_{t=1}^{\tau} \sum_{j \in [K]} \mathbb{I}\{a_t = j\}(\hat{r}_t(j) - \mu_j) \right] \\
&= \mathbb{E}\left[ \sum_{t=1}^{\tau} \sum_{j \in [K]} \mathbb{I}\{a_t = j\}(\hat{r}_t(j) - \mu_j) \Big| \mathcal{E}_C \right] \Pr(\mathcal{E}_C) \\
&\quad + \mathbb{E}\left[ \sum_{t=1}^{\tau} \sum_{j \in [K]} \mathbb{I}\{a_t = j\}(\hat{r}_t(j) - \mu_j) \Big| \bar{\mathcal{E}}_C \right] \Pr(\bar{\mathcal{E}}_C) \\
&\leq 2\tau/T + \mathbb{E}\left[ \sum_{t=1}^{\tau} \sum_{j \in [K]} \mathbb{I}\{a_t = j\}(\hat{r}_t(j) - \mu_j) \Big| \mathcal{E}_C \right] \\
&\leq 2\tau/T + 2 \left( \sigma \sqrt{d \log \left( \frac{1 + \tau L^2/\lambda}{\varrho} \right)} + \sqrt{\lambda} S \right) \mathbb{E}\left[ \sum_{t=1}^{\tau} \min \left\{ 1, ||x_{a_t}||_{\Sigma_{t-1}^{-1}} \right\} \Big| \mathcal{E}_C \right] \\
&\leq 2\tau/T + 2 \left( \sigma \sqrt{d \log \left( \frac{1 + \tau L^2/\lambda}{\varrho} \right)} + \sqrt{\lambda} S \right) \mathbb{E}\left[ \sqrt{\tau \sum_{t=1}^{\tau} \min \left\{ 1, ||x_{a_t}||_{\Sigma_{t-1}^{-1}}^2 \right\}} \Big| \mathcal{E}_C \right]
\end{aligned}
$$

where the last step is due to the Cauchy-Schwarz inequality. Using Lemma 11 in (Abbasi-Yadkori et al., 2011), we further have that

$$
\begin{aligned}
&\mathbb{E}\left[ \sum_{t=1}^{\tau} \sum_{j \in [K]} \mathbb{I}\{a_t = j\}(\hat{r}_t(j) - \mu_j) \right] \\
&\leq 2\tau/T + 2 \left( \sigma \sqrt{d \log \left( \frac{1 + \tau L^2/\lambda}{\varrho} \right)} + \sqrt{\lambda} S \right) \mathbb{E}\left[ \sqrt{2\tau \log \left( \frac{\det(\Sigma_\tau)}{\det(\lambda I_d)} \right)} \Big| \mathcal{E}_C \right] \\
&\leq 2\tau/T + 2 \left( \sigma \sqrt{d \log[T(1 + \tau L^2/\lambda)]} + \sqrt{\lambda} S \right) \sqrt{2\tau \log \left( 1 + \frac{\tau L^2}{\lambda d} \right)}.
\end{aligned}
$$

Similarly, we can derive an upper bound for this term

$$\mathbb{E}\Big[\sum_{t=1}^{\tau}\sum_{j\in[K]}\pi^*_{\epsilon_t,\delta}(j)(\mu_j-\hat{r}_t(j))\Big]$$

$$=\mathbb{E}\Big[\sum_{t=1}^{\tau}\sum_{j\in[K]}\pi^*_{\epsilon_t,\delta}(j)(\mu_j-\hat{r}_t(j))\Big|\mathcal{E}_C\Big]\Pr(\mathcal{E}_C)$$

$$+\mathbb{E}\Big[\sum_{t=1}^{\tau}\sum_{j\in[K]}\pi^*_{\epsilon_t,\delta}(j)(\mu_j-\hat{r}_t(j))\Big|\bar{\mathcal{E}}_C\Big]\Pr(\bar{\mathcal{E}}_C)$$

$$\leq 2\tau/T,$$

since under the event $\mathcal{E}_C$, we have that $\mu_j-\hat{r}_t(j)\leq 0$.

By setting $V_t=\delta\sqrt{t}, \epsilon_t=4\sqrt{1/t}$, we obtain the final upper bound for the regret. Recall that

$$\mathcal{T}_1(\tau)=\sum_{t=1}^{\tau}\sum_{j\in[K]}(\pi^*_j-\pi^*_{\epsilon_t,\delta}(j))\mu_j$$

$$\leq\sum_{t=1}^{\tau}\frac{\min\{\epsilon_t,\delta\}}{\delta}M$$

$$=\sum_{t=1}^{\lfloor\frac{16}{\delta^2}\rfloor}\frac{\min\{\epsilon_t,\delta\}}{\delta}M+\sum_{t=\lceil\frac{16}{\delta^2}\rceil}^{\tau}\frac{\min\{\epsilon_t,\delta\}}{\delta}M.$$

Observing that $\epsilon_t\geq\delta$ for $t\leq\lfloor\frac{16}{\delta^2}\rfloor$ and $\epsilon_t\leq\delta$ otherwise, we have

$$\mathcal{T}_1(\tau)\leq\sum_{t=1}^{\lfloor\frac{16}{\delta^2}\rfloor}\frac{\delta}{\delta}M+\sum_{t=\lceil\frac{16}{\delta^2}\rceil}^{\tau}\frac{\epsilon_t}{\delta}M$$

$$\leq\lfloor\frac{16}{\delta^2}\rfloor M+\frac{4M}{\delta}\int_{\lfloor\frac{16}{\delta^2}\rfloor}^{\tau}\frac{1}{\sqrt{x}}dx$$

$$=\lfloor\frac{16}{\delta^2}\rfloor M+\frac{8M}{\delta}\Big(\sqrt{\tau}-\sqrt{\lfloor\frac{16}{\delta^2}\rfloor}\Big).$$

Then, for $\mathcal{T}_2(\tau)$, we have that

$$\mathcal{T}_2(\tau)\leq\sum_{t=1}^{\tau}\frac{\bar{Z}^2+\epsilon_t^2}{V_t}+\sum_{t=1}^{\tau}\frac{t(\bar{Z}+\epsilon_1)(\epsilon_t-\delta)\mathbb{I}\{\epsilon_t>\delta\}}{V_t}$$

$$\leq\frac{\bar{Z}^2}{\delta}\sum_{t=1}^{\tau}\frac{1}{\sqrt{t}}+\frac{16}{\delta}\sum_{t=1}^{\tau}t^{-\frac{3}{2}}+(\bar{Z}+\epsilon_1)\sum_{t=1}^{\min\{\tau,\lfloor 16/\delta^2\rfloor\}}\frac{t\epsilon_t}{V_t}$$

$$\leq\frac{\bar{Z}^2}{\delta}\Big[1+\int_1^{\tau}\frac{1}{\sqrt{x}}dx\Big]+\frac{16}{\delta}\Big[1+\int_1^{\tau}x^{-\frac{3}{2}}dx\Big]+(\bar{Z}+\epsilon_1)\sum_{t=1}^{\min\{\tau,\lfloor 16/\delta^2\rfloor\}}\frac{t\epsilon_t}{V_t}$$

$$\leq\frac{\bar{Z}^2}{\delta}(2\sqrt{\tau}-1)+\frac{16}{\delta}(3-\frac{2}{\sqrt{\tau}})+(\bar{Z}+\epsilon_1)\frac{4}{\delta}\min\{\tau,\lfloor\frac{16}{\delta^2}\rfloor\}.$$

As a consequence, we observe that $\mathcal{R}(T)=\sum_{i\in[4]}\mathcal{T}_i(T)=\tilde{O}\big(\frac{\sqrt{T}}{\delta}+\frac{1}{\delta^3}\big)$.

### D.2. Cost Analysis Results

In this part, we analyze the constraint violation of the algorithm. The key is to prove an upper bound for $\mathbb{E}[L(t+1)-L(t)|\mathcal{H}_t]$, which implies an upper bound for $\mathbb{E}[||Q(t+1)||_2-||Q(t)||_2|\mathcal{H}_t]$. Then, by applying Lemma 11 in (Liu et al., 2021), we

are able to derive an upper bound for the constraint violation.

The constraint violation metric is defined as

$$Vio(\tau) = \mathbb{E}\Big[\sum_{t=1}^{\tau}\sum_{j\in[K]} Z_t(j)\mathbb{I}\{a_t = j\}\Big] \vee 0.$$

We rewrite the formula using our definition $Q(t)$. Note that

$$Q(t+1) = \Big[Q(t) + \sum_{j\in[K]} Z_t(j)\mathbb{I}\{a_t = j\} + \epsilon_t\Big]^+$$
$$\geq Q(t) + \sum_{j\in[K]} Z_t(j)\mathbb{I}\{a_t = j\} + \epsilon_t.$$

Thus, we can decompose $Q(\tau + 1)$ such that

$$Q(\tau+1) = Q(\tau+1) - Q(\tau) + Q(\tau) - Q(\tau-1) + \dots + Q(1) - Q(0) + Q(0)$$
$$\geq \sum_{t=1}^{\tau}\Big[\sum_{j\in[K]} Z_t(j)\mathbb{I}\{a_t = j\} + \epsilon_t\Big]$$
$$= \sum_{t=1}^{\tau}\sum_{j\in[K]} Z_t(j)\mathbb{I}\{a_t = j\} + \sum_{t=1}^{\tau}\epsilon_t.$$

Using this fact, we know that controlling $Vio(\tau)$ suffices to derive an upper bound for $Q(\tau + 1)$, because

$$Vio(\tau) \leq \mathbb{E}\Big[Q(\tau+1) - \sum_{t=1}^{\tau}\epsilon_t\Big] \vee 0 = \max\Big\{0, \quad \mathbb{E}[Q(\tau+1)] - \sum_{t=1}^{\tau}\epsilon_t\Big\}.$$

We derive an upper bound for $\mathbb{E}[Q(\tau + 1)]$ by showing that when the value of $Q$ is sufficiently large, it has a significantly negative expected shift in the next step, i.e.,

$$\mathbb{E}\Big[Q(t+1) - Q(t)\Big|\mathcal{H}_t\Big]$$
$$= \mathbb{E}\Big[\sqrt{Q(t+1)^2} - \sqrt{Q(t)^2}\Big|\mathcal{H}_t\Big]$$
$$\leq \frac{1}{2Q(t)}\mathbb{E}\Big[Q(t+1)^2 - Q(t)^2\Big|\mathcal{H}_t\Big]$$
$$= \frac{1}{2Q(t)}\mathbb{E}\Big[L(t+1) - L(t)\Big|\mathcal{H}_t\Big].$$

The inequality is due to the concavity of the function $\sqrt{\cdot}$. Recall our upper bound on the Lyapunov drift, selecting the policy to be $\pi_\delta$ defined in Assumption 4.6,

$$\mathbb{E}\Big[L(t+1) - L(t)\Big|\mathcal{H}_t\Big] \leq (\bar{Z}^2 + \epsilon_t^2) + Q(t)\mathbb{E}\Big[\sum_{j\in[K]} Z_t(j)\pi_\delta(j) + \epsilon_t\Big|\mathcal{H}_t\Big]$$
$$- V_t\mathbb{E}\Big[\sum_{j\in[K]}(\pi_\delta(j) - \mathbb{I}\{a_t = j\})\hat{r}_t(j)\Big|\mathcal{H}_t\Big]$$
$$\leq (\bar{Z}^2 + \epsilon_t^2) + Q(t)\sum_{j\in[K]} c_j\pi_\delta(j) + Q(t)\epsilon_t$$
$$- V_t\mathbb{E}\Big[\sum_{j\in[K]}(\pi_\delta(j) - \mathbb{I}\{a_t = j\})\hat{r}_t(j)\Big|\mathcal{H}_t\Big]$$
$$\leq (\bar{Z}^2 + \epsilon_t^2) + Q(t)(\epsilon_t - \delta) - V_t\mathbb{E}\Big[\sum_{j\in[K]}(\pi_\delta(j) - \mathbb{I}\{a_t = j\})\hat{r}_t(j)\Big|\mathcal{H}_t\Big].$$

Note that $\hat{r}_t(j)$ is clipped into $[-1, 1]$, which means that

$$\sum_{j \in [K]} \mathbb{I}\{a_t = j\}\hat{r}_t(j) \leq \sum_{j \in [K]} \mathbb{I}\{a_t = j\} = 1$$

$$\sum_{j \in [K]} \pi_\delta(j)\hat{r}_t(j) \geq \sum_{j \in [K]} \pi_\delta(j)(-1) = -1$$

and of course

$$\mathbb{E}\Big[L(t+1) - L(t)\Big|\mathcal{H}_t\Big] \leq (\bar{Z}^2 + \epsilon_t^2) + Q(t)(\epsilon_t - \delta) + 2V_t.$$

If $\epsilon_t \leq \frac{\delta}{2}$ and $Q(t) \geq \phi(t) := \frac{4(2V_t + \bar{Z}^2 + \epsilon_t^2)}{\delta}$,

$$\mathbb{E}\Big[Q(t+1) - Q(t)\Big|\mathcal{H}_t\Big]$$
$$\leq \frac{1}{2Q(t)}\Big[(\bar{Z}^2 + \epsilon_t^2) - \frac{\delta}{2}Q(t) + 2V_t\Big]$$
$$= \frac{2V_t + \bar{Z}^2 + \epsilon_t^2}{2Q(t)} - \frac{\delta}{4}$$
$$\leq \frac{2V_t + \bar{Z}^2 + \epsilon_t^2}{2\phi(t)} - \frac{\delta}{4} = -\frac{\delta}{8}.$$

Lemma 11 in (Liu et al., 2021) implies that $\forall t_0 : \epsilon_{t_0} \leq \frac{\delta}{2}$,

$$\mathbb{E}\Big[\exp(\xi Q(t))\Big] \leq \exp(\xi Q(t_0)) + \frac{16}{\xi\delta} \exp\Big(\xi(\bar{Z} + \epsilon_1 + \phi(t))\Big), \quad \xi = \frac{\delta/8}{(\bar{Z} + \epsilon_1)^2 + (\bar{Z} + \epsilon_1)\delta/24}.$$

Recall our objective, for $t_0$ such that $\epsilon_{t_0} \leq \frac{\delta}{2}$:

$$\mathbb{E}[Q(t)] = \mathbb{E}\Big[\frac{\xi Q(t)}{\xi}\Big]$$
$$= \frac{1}{\xi}\mathbb{E}\Big[\log\exp(\xi Q(t))\Big]$$
$$\leq \frac{1}{\xi}\log\mathbb{E}\Big[\exp(\xi Q(t))\Big]$$
$$\leq \frac{1}{\xi}\log\Big(\exp(\xi Q(t_0)) + \frac{16}{\xi\delta}\exp\Big(\xi(\bar{Z} + \epsilon_1 + \phi(t))\Big)\Big)$$
$$\leq \frac{1}{\xi}\log\Big(\exp\Big(\xi(Q(t_0) + \bar{Z} + \epsilon_1 + \phi(t))\Big) + \frac{16}{\xi\delta}\exp\Big(\xi(Q(t_0) + \bar{Z} + \epsilon_1 + \phi(t))\Big)\Big)$$
$$= \frac{1}{\xi}\log\Big(1 + \frac{16}{\xi\delta}\Big) + Q(t_0) + \bar{Z} + \epsilon_1 + \phi(t)$$
$$\leq \frac{1}{\xi}\log\Big(1 + \frac{16}{\xi\delta}\Big) + (1 + t_0)\bar{Z} + \sum_{t=1}^{t_0}\epsilon_t + \epsilon_1 + \phi(t).$$

By selecting $V_t = \delta\sqrt{t}, \epsilon_t = 4\sqrt{1/t}$, we have the final upper bound

$$
Vio(\tau) \leq \max\left\{0, \mathbb{E}[Q(\tau+1)] - \sum_{t=1}^{\tau}\epsilon_t\right\}
$$

$$
\leq \max\left\{0, \frac{1}{\xi}\log\left(1 + \frac{16}{\xi\delta}\right) + (1+t_0)\bar{Z} + \sum_{t=1}^{t_0}\epsilon_t + \epsilon_1 + \frac{4(2V_{\tau+1} + \bar{Z}^2 + \epsilon_{\tau+1}^2)}{\delta} - \sum_{t=1}^{\tau}\epsilon_t\right\}
$$

$$
= \max\left\{0, \frac{1}{\xi}\log\left(1 + \frac{16}{\xi\delta}\right) + (1+t_0)\bar{Z} + \sum_{t=1}^{t_0}\epsilon_t + \epsilon_1 + \frac{4(\bar{Z}^2 + \epsilon_{\tau+1}^2)}{\delta} + 8\sqrt{\tau+1} - \sum_{t=1}^{\tau}\epsilon_t\right\}
$$

$$
\leq \max\left\{0, \frac{1}{\xi}\log\left(1 + \frac{16}{\xi\delta}\right) + (1+t_0)\bar{Z} + \sum_{t=1}^{t_0}\epsilon_t + \epsilon_1 + \frac{4(\bar{Z}^2 + \epsilon_{\tau+1}^2)}{\delta} + 8\sqrt{\tau+1} - 4\int_1^{\tau+1}\frac{dx}{\sqrt{x}}\right\}
$$

$$
\leq \max\left\{0, \frac{1}{\xi}\log\left(1 + \frac{16}{\xi\delta}\right) + (1+t_0)\bar{Z} + \sum_{t=1}^{t_0}\epsilon_t + \epsilon_1 + \frac{4(\bar{Z}^2 + \epsilon_1^2)}{\delta} + 8\right\}.
$$

Choosing $t_0 = \lceil\frac{64}{\delta^2}\rceil$, we have that

$$
Vio(\tau) \leq \max\left\{0, \frac{1}{\xi}\log\left(1 + \frac{16}{\xi\delta}\right) + (1+t_0)\bar{Z} + (1+t_0)\epsilon_1 + \frac{4(\bar{Z}^2 + \epsilon_1^2)}{\delta} + 8\right\}
$$

$$
\leq \max\left\{0, \frac{1}{\xi}\log\left(1 + \frac{16}{\xi\delta}\right) + (1+t_0)(\bar{Z} + \epsilon_1) + \frac{4(\bar{Z}^2 + \epsilon_1^2)}{\delta} + 8\right\}
$$

$$
= \frac{(\bar{Z}+4)^2 + (\bar{Z}+4)\delta/24}{\delta/8}\log\left(1 + \frac{16(\bar{Z}+4)^2 + 2(\bar{Z}+4)\delta/3}{\delta^2/8}\right)
$$

$$
+ \left(1 + \lceil\frac{64}{\delta^2}\rceil\right)(\bar{Z}+4) + \frac{4(\bar{Z}^2 + 16)}{\delta} + 8.
$$

