# OpenReview forum: "The Pareto-optimal Trade-off between Regret and Statistical Inference in Linear Stochastic Bandits under Safety Constraints"
_ICML.cc/2026/Conference — ICML 2026 regular_

### Official Review · Reviewer_shER · 2026-03-11

**Soundness:** 3
**Presentation:** 3
**Significance:** 3
**Originality:** 2
**Overall Recommendation:** 3
**Confidence:** 3

**Summary:**

This paper studies safety-constrained linear bandits with the goal of simultaneously controlling regret, parameter estimation error, and cumulative safety violation. The authors establish a minimax lower bound characterizing the trade-off between regret, safety, and statistical estimation accuracy. To match this frontier, they propose a two-phase algorithm SERMiSC that performs safety-aware exploration for parameter estimation in the first phase and switches to safe regret minimization in the second phase. Theoretical results show that the algorithm achieves sublinear regret, near-constant safety violation, and estimation error matching the lower bound up to logarithmic factors.

**Compliance With Llm Reviewing Policy:**

Affirmed.

**Final Justification:**

Thank you for the reply. If the framework adopts an expectation-based (soft) safety constraint rather than a hard constraint, I find the motivation of this modeling choice not fully convincing.

In particular, the proposed trade-off between regret minimization and parameter estimation error appears less compelling than suggested. In standard linear bandit settings, improved estimation accuracy typically leads to better decision-making and thus lower regret, making the tension between these two objectives less pronounced.

From this perspective, the problem formulation seems closer to incorporating a form of pure exploration into the linear bandit framework, where the trade-off primarily lies between additional exploration and short-term regret, rather than introducing a fundamentally new three-way trade-off involving safety (safety plays no constraining role in the framework).

Overall, while the framework is interesting, I find the conceptual motivation and distinctiveness of the proposed trade-off insufficiently justified. Based on these considerations, I believe the score is appropriate.

**Key Questions For Authors:**

1. The proposed algorithm consists of two phases, but the description of the budget allocation and stopping mechanism is not entirely clear to me. For example, is it possible that the safety cost budget is exhausted before reaching the horizon T? If that happens, how would the algorithm proceed? Also, how is the splitting point between the two phases determined? Is the length of the first phase a parameter that needs to be chosen in advance, and if so, are there any theoretical or practical guidelines for selecting it?
2. The paper analyzes the two phases separately, but in practice they seem closely coupled. In particular, the cost incurred in the first phase directly affects the remaining safety budget for the second phase, and the exploration behavior in the first phase may also influence the reward performance later on. It would be helpful if the authors could clarify how these two phases interact and how the behavior in the first phase affects the overall performance.
3. The experimental section is relatively brief. It would be helpful if the authors could provide more details about the experimental configuration and discuss how the algorithm performs in larger-scale or more realistic settings. The paper does not seem to provide a more systematic discussion of the trade-off between the length of the first phase and the total horizon T, nor does it provide guidance on how this parameter should be chosen in practice.
4. The algorithm involves several parameters. Some discussion on parameter sensitivity would help better understand the robustness of the proposed algorithm.

**Limitations:**

Yes

**Strengths And Weaknesses:**

Strengths
1. Studies an interesting problem that jointly considers regret minimization, parameter estimation accuracy, and safety constraints in linear bandits.
2. Provides a theoretical framework characterizing the trade-off via a minimax lower bound and a matching algorithm.
3. The paper is generally well structured, with a clear connection between the lower bound and the two-phase algorithm design.

Weaknesses
1. Some theoretical components (e.g., the lower-bound construction and policy class assumptions) could benefit from clearer explanation.
2. The empirical evaluation is limited and mainly conducted on small-scale synthetic experiments.
3. The algorithm mainly combines existing ideas from safe bandits and constrained optimization, and the algorithmic novelty is moderate.

---

> ### Author Rebuttal · Authors · 2026-03-30
>
> W:
>
> 1. We thank the reviewer and commit to expanding the exposition of the lower bound and policy class assumptions in the revision.
>
> 2. Our work is primarily theoretical: our main contributions are the minimax lower bound characterizing the Pareto-optimal frontier and a new algorithm provably matches it. In line with common practice in the bandit theory literature, we include synthetic experiments to validate and illustrate our theoretical guarantees.
>
> 3. We respectfully note that the primary focus of this paper is not algorithm design but rather the characterization of fundamental limits: Thm 3.1 rigorously characterizes the Pareto-optimal frontier of the regret–inference–safety trade-off. The two-phase algorithm is intentionally designed to be simple and conceptually aligned with the lower bound, demonstrating its tightness. In this context, algorithmic simplicity is a feature rather than a limitation.
>
> Q:
>
> 1. (1) **Budget exhaustion.** This scenario does not arise in our setup. The safety constraint violation functions as a performance metric rather than a hard budget: its value does not trigger any stopping condition or alter the algorithm's execution flow.
>
>     (2) We thank the reviewer for raising this point. $T_1$ is a pre-specified parameter, and clear guidelines for selecting it are already provided in our paper. Specifically, Table 1 summarizes explicitly how $T_1$ governs all three metrics: increasing $T_1$ tightens the estimation error bound at the cost of higher regret, and vice versa. In practice, the practitioner can therefore choose $T_1$ according to their preference over the regret--inference frontier.
>
> 2. We appreciate this question and clarify the inter-phase relationship along two dimensions:
>
>     **Safety cost.** The two phases are indeed coupled in this respect: the total constraint violation at any time $t>T_1$ is the sum of costs accumulated across both phases. This is precisely why Table 1 reports the total safety violation $\tilde{O}(1)$, a bound that already accounts for this accumulation.
>
>     **Estimation and reward performance.** The two phases are in fact decoupled here, which is a deliberate design choice. Phase II discards Phase I's estimator and constructs a fresh one using only data collected from $t=T_1+1$ onward for the purpose of regret minimization. Consequently, the exploration behavior in Phase I does not affect Phase II's reward performance. The only channel through which Phase I influences overall performance is through its duration $T_1$: as summarized in Table 1, a longer Phase I yields a smaller estimation error but contributes a larger $O(T_1)$ term to the total regret.  We refer the reviewer to Table 1 for a consolidated view of how $T_1$ propagates through both phases to determine the algorithm's end-to-end guarantees.
>
> 3. We acknowledge that the experimental section could benefit from additional implementation details, and we commit to including a dedicated configuration description in the appendix of the revised version.
>
>     Regarding the systematic discussion of the trade-off between $T_1$ and $T$, we respectfully note that this is already addressed in the main text. Table 1 provides a consolidated characterization, which directly yields practical guidance: a practitioner can select $\beta\in [1/2, 1)$ according to their application's priorities ($T_1:=\lfloor T^\beta \rfloor$). We believe the theoretical guarantees in Table 1, combined with the ablation in Figure 1, offer guidance that is both rigorous and actionable. That said, we are happy to add a brief summarizing remark in the experimental section to make this connection more explicit for the reader.
>
> 4. We appreciate this question, which we believe arises from the common experience with Machine Learning systems where hyperparameter sensitivity is difficult to characterize theoretically. Our setting is fundamentally different in this regard.
>
>     The parameters in SERMiSC fall into two categories, both of which enjoy clear theoretical justification. First, parameters such as $\zeta_t, V_t,$ and $D$ are derived from the theoretical analysis: their specific forms are dictated by the proof of our upper bound theorems, and are not free choices subject to empirical tuning. Second, parameters such as $\lambda$ and $T_1$ appear explicitly in the theoretical bounds themselves, so their effect on performance is fully transparent, e.g., a practitioner can read off directly from Table 1 and the bound expressions how these parameters influence the regret - estimation trade-off, and select them accordingly.
>
>     In summary, the notion of parameter sensitivity as typically understood in empirical machine learning, where one must search over configurations without theoretical guidance, does not apply here. Every parameter either has a theoretically optimal specification or appears transparently in the performance bounds, giving users principled and interpretable control over the algorithm's behavior.

---

> > ### Author Rebuttal · Reviewer_shER · 2026-04-02
> >
> > I thank the authors for the detailed rebuttal, which helps clarify several of my concerns. However, some questions are only partially addressed.
> >
> > In particular, regarding Q1, while safety violation is treated as a performance metric rather than a hard constraint, it remains unclear why this rules out scenarios with excessive violations, and how the algorithm is guaranteed to run until the full horizon T. It is also not fully clear how infeasible policies are characterized or handled, which makes the role of the safety constraint less explicit.
> >
> > Overall, I keep my score unchanged.

---

> > > ### Author Response · Authors · 2026-04-04
> > >
> > > Q: 'it remains unclear why this rules... the full horizon T.'
> > >
> > > We thank the reviewer for this thoughtful follow-up, which helps us identify a point of miscommunication in our previous response. We described the safety cost as a "performance metric," which may have inadvertently suggested that safety plays no constraining role in our framework. We appreciate the opportunity to clarify and restate our position more precisely.
> > >
> > > Our framework adopts an **expectation-based (soft) safety constraint**, following the formulation of [1, 2]. We believe the reviewer's concern stems from interpreting "constraint" as necessarily implying hard, runtime enforcement — e.g., the algorithm must terminate once cumulative violation exceeds a threshold. Under such a hard-constraint model, the question of budget exhaustion before horizon $T$ is indeed natural. However, our soft-constraint formulation operates differently: the safety requirement $\mathcal{C}(\tau)\leq O (\delta_T)$ constrains the **design** of the algorithm (i.e., we seek algorithms whose expected cumulative cost remains effectively controlled), but does not impose any runtime enforcement mechanism such as stopping. The algorithm executes for all $T$ rounds by construction.
> > >
> > > This modeling choice is a deliberate simplification, not an oversight. In practice, the relationship between a hard constraint and algorithm behavior is direct and unambiguous: once the budget is exhausted, execution halts. By contrast, the interaction between a soft constraint and runtime behavior is far more nuanced — it depends heavily on the specific enforcement mechanism one chooses to model (e.g., absorbing-state termination, penalty escalation, frequency-based triggers), each of which leads to a fundamentally different algorithmic and analytical framework. Our soft-constraint formulation abstracts away from these implementation-specific details to isolate the fundamental Pareto trade-off structure among regret, inference, and safety — which is the central contribution of this work. We note that this same modeling philosophy underlies [1, 2] and is standard in the constrained bandit literature. We recognize that incorporating explicit stopping rules into the safety-constrained bandit framework is a valuable direction for future work.
> > >
> > > That said, we emphasize that "soft" does not mean "weak." SERMiSC provides **anytime safety guarantees** that go well beyond what the terminal soft-constraint formulation requires. It can be verified that the total safety cost satisfies $\mathcal{C}(\tau) = O(\tau^{o(1)})$ for all $\tau\in[T]$. This means that even if a hard stopping rule were externally imposed, SERMiSC would remain safe at any interruption point. In summary, SERMiSC rules out scenarios with excessive violations not through a runtime enforcement mechanism, but through its algorithmic design and provable anytime safety guarantees.
> > >
> > >
> > >
> > >
> > > Q: 'It is also not fully clear... less explicit.'
> > >
> > > We believe this concern also originates from the hard-vs-soft distinction discussed above. Under a hard-constraint formulation, "infeasible policy" is a well-defined concept — a policy that violates the constraint cannot be executed or must be terminated. Under our soft-constraint formulation, however, there are no infeasible policies: every policy $\pi$ is executable for the full horizon, and the safety cost $\mathcal{C}(\tau)$ is a quantity that different policies achieve to different degrees. The question of how to "handle" infeasible policies therefore does not arise in our model.
> > >
> > > That said, the safety constraint does play an explicit and substantive role in our framework. Theoretically, it shapes the Pareto frontier: the lower bound in Theorem 3.1 shows that tighter safety tolerance (smaller $\delta$) raises the estimation error floor, making the trade-off more stringent. Practically, it constrains algorithm design: we prefer algorithms whose safety violation remains at the $\tilde{O}(1)$ level.
> > >
> > > We hope the above clarifications adequately address the reviewer's concerns. Should any questions remain, we would be very happy to continue the discussion and provide further details. If the reviewer finds these clarifications satisfactory and the concerns have been resolved, we would respectfully ask whether the reviewer might consider re-evaluating the score accordingly. We sincerely appreciate the reviewer's time and constructive engagement throughout this discussion.
> > >
> > >
> > > [1] Liu et al. An Efficient Pessimistic-Optimistic Algorithm for Stochastic Linear Bandits with General Constraints. 2021
> > >
> > > [2] Ma et al. High-dimensional linear bandits with knapsacks. 2023

---

### Official Review · Reviewer_mao9 · 2026-03-13

**Soundness:** 2
**Presentation:** 2
**Significance:** 2
**Originality:** 2
**Overall Recommendation:** 4
**Confidence:** 3

**Summary:**

This paper studied the problem of stochastic linear bandit, with a multi-objective of regret and estimation error, under some linear (called "safety") constraint. Authors proved some Pareto minmax lower bound and proposed an algorithm that is Pareto optimal.

**Compliance With Llm Reviewing Policy:**

Affirmed.

**Final Justification:**

I appreciate further responses by authors and I believe they will improve on the writing of the paper. Still, after reading all other reviews, I am a bit suspicious on the motivation of the framework as it may not introduce inherent challenges over existing ones, as well as the technical contribution of the paper. However, suppose all the analysis are correct, and given that the setting seems new and authors promised to improve the writing for the final submission, I would like to raise my score to 4 temporarily. I will certainly do a more detailed investigation regarding the novelty of the theoretical results these days.

**Key Questions For Authors:**

All my questions are listed in **Weaknesses** above.

**Limitations:**

Should mention the limitations of the assumptions made throughout.

**Strengths And Weaknesses:**

**Strengths:**
1. The setup of studying multiple objectives of regret, estimation error and safety violation seems novel.
2. The Pareto minimax lower bound and the two-phase algorithm seem interesting and non-trivial.

**Weaknesses:**
1. Needs clarification on the expression of $C(\tau)$ on page 2. Say, what is $\tau$? Should unify the notation for horizon. And why does this constraint make sense? Needs more contexts on this safety constraint.
2. What's vector $\mathbf{c}$ in the definition of $P(\gamma)$ on page 3? If that's related to safety constraint defined on page 2, then it's better to write it into the expression of $C(\tau)$.
3. Need more contexts on why the family $\Pi(\beta, \delta)$ is defined that way. Also, why introduce a sequence of $\delta_t$ when only the last $\delta_T$ matters in the result?
4. Questionable writings in Theorem 4.1: when $\underline q$ also depends on $T_1$, it's unclear why the bound is $O(T_1^{-1/2})$. Also, does the bound hold for any choice of $\zeta_t$?
5. Update in Algorithm 1 done wrong in the for loop.
6. Lack of justification of assumptions, say Assumption 4.2 and 4.5.
7. When there's always safety violation, why requiring $C(\tau)\leq 0$ instead of taking $C(\tau)$ as another objective that should be minimized? More specifically, why not defining a new objective as the sum of cumulative regret and cumulative safety violation, along with the estimation error alone as the second objective, as suggested in the Pareto minimax lower bound in equation (2)?

---

> ### Author Rebuttal · Authors · 2026-03-30
>
> W:
> 1. We respectfully clarify that $T$ is reserved for the total time horizon, while $\tau\in[T]$ is a generic intermediate time index. We will add an explicit qualifier at the first appearance of $\tau$ in the revision.
>
>     Our safety constraint $\mathcal{C}(\tau)$ is an anytime cumulative constraint [1]. Similar cumulative safety constraints are widely adopted in the literature [2-3], where the signed cost enforces a net balance condition. We provide these real-world examples to further justify its naturality. In clinical trials, it encodes a net non-maleficence requirement: the expected cumulative harm must remain non-positive, a common ethical standard. In online recommendation, it prevents sustained harm to user experience. In network scheduling, it ensures no net resource deficit accumulates. Across these domains, safety is monitored at the temporal-average level via expected outcomes, making the expectation-based formulation a faithful reflection of real-world practice.
>
>
> 2. We thank the reviewer for this observation. The vector $\textbf{c}=(c_a)_{a\in[K]}$ collects the expected costs already introduced in Line 092, and is also listed as an example of boldface vector notation in Line 108. We will nonetheless add a clarification in the revision.
>
>
> 3. The policy family $\Pi(\beta, \delta)$ is defined precisely to serve Thm 3.1: it characterizes all algorithms simultaneously achieving prescribed regret and safety levels, parameterized by $\beta$ and $\delta$. Intuitively, the more stringently an algorithm controls regret and safety violations, the less freedom it has to explore, and consequently the worse its estimation accuracy must be. The definition of $\Pi$ is therefore the natural domain over which the Pareto-optimal trade-off is characterized.
>
>     The sequence $\{\delta_t\}$ is introduced to conceptually accommodate the anytime cumulative constraint paradigm, which requires safety to hold at each round $\tau$, not merely at $T$. The fact that only $\delta_T$ appears in bound (2) is a consequence of a relaxation in the KL divergence upper bound. Formally, consider a strictly larger class $\Pi'(\beta, \delta)$ requiring only the terminal condition $\mathcal{C}(T)=O(\delta_T)$. Since $\Pi\subseteq\Pi'$, the same lower bound holds over $\Pi'$. The use of $\{\delta_t\}$ thus makes the theorem more broadly applicable: it is compatible with the anytime cumulative paradigm while the conclusion remains valid even under the terminal-only requirement. We will add a clarifying remark in the revision.
>
> 4. $\underline{q}$ denotes a positive lower bound on the user-specified hyperparameter sequence $\{q_t\}$, which is a property of the algorithm's design rather than a quantity that scales with $T_1$. Any reasonable choice, e.g., the constant sequence $q_t$ in Corollary 4.4, trivially admits a constant $\underline{q}$ independent of $T_1$, making the asymptotic rate unambiguous. We will restate Thm 4.1 by explicitly assuming the existence of such a constant in the revision.
>
>     Thm 4.1 holds for any choice of $\zeta_t$. As seen in the proof, $\zeta_t$ does not appear in the estimation error analysis and only affects the safety cost analysis in Thm 4.3. The specific choice in (4) is what guarantees the safety violation bound in Thm 4.3, while leaving the estimation bound of Thm 4.1 unaffected.
>
> 5. We thank the reviewer for this comment, but respectfully maintain that the updates in Algo 1 are correct. $Y_a$ and $N_a$ accumulate the raw rewards and pull counts across Phase I. After the loop, $Y_a/\sqrt{N_a}$ and the design matrix with rows $\sqrt{N_a}x_a^\top$ together recover exactly the standard OLS estimator.
>
> 6. Asp 4.2: The almost sure boundedness of $Z_t(a)$ is standard in the constrained bandit literature, and $|c_{\min}| \geq \varsigma \bar{c}$ is a mild non-degeneracy condition ensuring that conservative arms provide a meaningful safety buffer.
>
>     Asp 4.5: Slater's condition is a standard constraint qualification widely adopted in the constrained bandit literature.
>
>     We'll add justifications in the revision.
>
> 7. We thank the reviewer for this suggestion. Treating safety violation as a constraint rather than collapsing it into a weighted sum with regret is the standard modeling choice in the constrained bandit literature [1-3], reflecting the conceptual distinction between safety as a hard requirement and regret as a performance objective. Moreover, the suggested formulation is closely related to Lagrangian duality, which is precisely the technique we employ in both the lower bound proof and the Phase II design. We believe the two formulations are closely related and would likely yield similar analytic results.
>
> [1] Liu et al. An Efficient Pessimistic-Optimistic Algorithm for Stochastic Linear Bandits with General Constraints. 2021
>
> [2] Liu et al. Combinatorial Bandits with Linear Constraints: Beyond Knapsacks and Fairness. 2022
>
> [3] Ma et al. High-dimensional linear bandits with knapsacks. 2023

---

> > ### Author Rebuttal · Reviewer_mao9 · 2026-04-04
> >
> > I thank authors for detailed explanations and some of my concerns are resolved. However, I am still confused about the statements and writing of several results. For example, Theorem 4.1 is not self-contained, as reading it alone won't make me understand why the bound is $O(T_1^{-1/2})$. If this can be achieved by taking specific values of $q_t$, then it's better to make it clear in the statement of the theorem. Also Theorem 3.1 seems a bit weak to me if your class of policies depends on a sequence of $\delta_t$ while your lower bound only depends on $\delta_T$. If getting a $(\delta_t)_t$-dependent lower bound is too hard, then why defining the class to be $(\delta_t)_t$-dependent in the first place?
> >
> > Based on above arguments, I keep my score at this point.

---

> > > ### Author Response · Authors · 2026-04-05
> > >
> > > We sincerely thank the reviewer for the continued engagement and for acknowledging that some of the earlier concerns have been resolved. We appreciate the reviewer's specific suggestions on improving the clarity of our theorem statements, and we address each point below.
> > >
> > > **On Theorem 4.1.** We agree with the reviewer that the current statement of Theorem 4.1 is not self-contained, as reading it alone does not make the $O(T_1^{-1/2})$ rate immediately apparent. As we explained in our previous response, $\underline{q}$ is a strictly positive constant determined by the algorithm's hyperparameter choice and does not scale with $T_1$, but this should be made explicit within the theorem statement itself rather than requiring the reader to consult surrounding discussion. In the revised manuscript, we will restate Theorem 4.1 by either incorporating a specific choice of $\{q_t\}$ (e.g., $q_t = \frac{\varsigma}{1+\varsigma}$ as in Corollary 4.4) directly into the theorem, or by explicitly assuming the existence of a constant $\underline{q}>0$ independent of $T_1$ as a precondition, so that the $O(T_1^{-1/2})$ rate is unambiguous from the theorem statement alone.
> > >
> > > **On Theorem 3.1 and $\{ \delta_t \}$.**  The reviewer's observation is valid: defining $\Pi(\beta, \delta)$ with the full sequence $\{ \delta_t \}$ while the lower bound depends only on $\delta_T$ does make the result appear weaker than necessary. **Our original motivation for introducing the sequence was conceptual — to align the policy class with the anytime safety constraint paradigm**, where the safety requirement is respected at every intermediate round rather than merely at the terminal time. However, we agree that this comes at the cost of clarity and conciseness in the theorem statement.
> > >
> > > Since $\Pi(\beta, \delta)\subseteq \Pi'(\beta, \delta_T)$, where $\Pi'$ imposes only the terminal condition $\mathcal{C}(T)=O(\delta_T)$, the same lower bound holds over the larger class $\Pi'$, yielding a strictly stronger conclusion. In the revised manuscript, we will restate Theorem 3.1 using the simpler class $\Pi'(\beta, \delta_T)$ as the primary result, and add a remark noting that the bound remains valid under the more restrictive anytime formulation.
> > >
> > > We hope these revisions address the reviewer's remaining concerns on clarity and self-containedness. We believe the underlying theoretical contributions, the Pareto minimax lower bound and the matching algorithm, are sound, and that the issues raised in this round are presentational in nature, which we are committed to fixing in the revised manuscript. Should any further questions remain, we are happy to continue the discussion. If the reviewer finds these concerns resolved, we would be grateful if the reviewer might consider re-evaluating the score. We sincerely appreciate the reviewer's constructive feedback, which has helped us improve the presentation of this work.

---

### Official Review · Reviewer_yy9W · 2026-03-22

**Soundness:** 3
**Presentation:** 3
**Significance:** 2
**Originality:** 2
**Overall Recommendation:** 4
**Confidence:** 4

**Summary:**

This paper studies linear bandits with safety constraints and two competing objectives: parameter estimation and regret minimization. The main contributions are:
1. a lower bound, characterizing the achievable l2 estimation error given bounds on regret and safety constraint violation,
2. a two-phase algorithm with a matching upper bound,
3. a synthetic simulation study, comparing the proposed approach to two baselines.

**Compliance With Llm Reviewing Policy:**

Affirmed.

**Final Justification:**

The authors clarified my questions and while I can recommend acceptance I keep my score as explained in the comment to the authors.

**Key Questions For Authors:**

How is this setting different from minimizing estimation error subject to two constraints, one safety constraint and a bound on the regret? Why do we need to treat regret separately? Couldn't we generalize the setting to 2+ constraints, and treat regret this way?

**Limitations:**

Limitations are not discussed in the paper.

The paper is theory-focused and there is no direct potential negative societal impact.

**Strengths And Weaknesses:**

## Strengths

* The paper presents an analysis of the linear bandit setting with a novel objective that aims to minimize regret, while simultaneously guaranteeing a bound on the parameter estimation error and satisfy a cumulative safety constraint.
* The lower bounds charaterizes the trade-off between regret, safety constraints and achievable estimation error.
* The proposed algorithm is a relatively simple two-phase algorithm, where in the first phase, the algorithm aims to minimize estimation error, subject to the safety constraint, and in the second phase, the algorithm minimizes regret subject to the safety constraint. The pareto trade-off appears by tuning the length of each phase.
* The analysis is sound as far as I can tell.
* The paper addresses an interesting setting that appears to be novel, and while the authors provide some motivation, the setting could be better motivated.
* Related work is discussed adequately.

## Weaknesses

* The setting and results should be better motivated: Why do we care about simultaneously minimizing regret and minimizing estimation error? What is the algorithmic implication? In particular, the solution is relatively simple: First estimate the parameter, suffering o(1) regret per step, and then minimizing regret while not directly improving the parameter estimate. The trade-off between the two is thus a direct consequence of the length of each phase. The main insight is perhaps that this trade-off is optimal.
* The main result needs to be formalized better: How is an "estimator" defined? Why is there an index set over \\(\nu\\), i.e. what is the range of \\(\max_{\nu}\\) ? Why not directly maximize over some set of parameters? (I can see why it was written this way by looking at the proof that follows, but the theorem statement is not self contained, and hard to understand without looking at the proof)
* Please formally state the complete upper bound of the algorithm. Right now the main text only provides statements for each of the phases, and a discussion in section 4.3.
* The estimation phase uses uniform estimation, this can likely be improved using a spanning set or optimal design, e.g. D-optimal design. Perhaps the authors could comment on how such an improvement affects the final bound? E.g. the dependency on the number of arms.
* The experimental evaluation is limited to a single synthetic instance and two baselines.

**Minors:**
* Line 201, second column: Should it be \tilde \pi insde the KL?
* Line 073, first column: What is ATE?

---

> ### Author Rebuttal · Authors · 2026-03-30
>
> W:
>
> 1. **Motivation.** Clinical trials illustrate the necessity: experimental data is expensive yet indispensable for inference about treatment mechanisms, enabling generalization under covariate shift — and must be collected during active patient treatment. Inference, regret, and safety are thus intrinsically coupled in clinical experimentation, yet the literature has largely studied them in isolation.
>
>     **Algorithmic novelty.** The reviewer's reading is accurate, and we agree: the central insight is that this trade-off is optimal. Our primary contribution is Thm 3.1, which characterizes the fundamental limits of the regret-inference-safety frontier. The two-phase algorithm is intentionally transparent so that it cleanly demonstrates tightness of the lower bound. Algorithmic simplicity here reflects the sharpness of the bound rather than a lack of analytical depth.
>
> 2. We thank the reviewer for this precise observation and agree that Thm 3.1 would benefit from a more self-contained statement. We will address each point in the revision.
>
>     Following the convention in the related literature [1], an estimator is formally a measurable mapping from the history to $\mathbb{R}^d$. The environment $\nu$ can be characterized by the pair $(\theta^*, \textbf{c})$, the parameter vector and cost vector, since the arm features $\mathcal{X}$ are fixed and known, and the subGaussian parameter $\sigma$ is a known input to the algorithm. Accordingly, $\max_{\nu}$ ranges over $\mathcal{B}_{\theta} \times C$, where $C$ is a feasible set of the cost vectors. We will rewrite the theorem with this explicit range, making the statement self-contained without reference to the proof.
>
> 3. Thanks for this suggestion. While Table 1 already consolidates the total guarantees, we agree a formal unified statement would strengthen the paper and commit to stating the end-to-end upper bounds as the sum across both phases.
>
> 4. This is an insightful observation. The choice of the exploratory policy $\pi'$ directly affects the constant factor in the estimation error bound of Thm 4.1. With the uniform distribution, the factor scales with $K$,
> as sampling budget is spread evenly across all arms including redundant ones. A better strategy would concentrate probability mass on a spanning set of size $d$, a significant improvement when $d \ll K$.
>
>     Fixing $q_t$ as in Corollary 4.4, the special choice $q_t = \varsigma / (1+\varsigma)$ guarantees that the safety constraint $\sum_{i=1}^K c_i\pi'(i)\sum_{t=1}^{T_1}q_t + c_{\min} \sum_{t=1}^{T_1}(1-q_t)\leq 0$ is automatically satisfied for any $\pi'$ under Asp 4.2, without requiring knowledge of $\textbf{c}$. The optimal $\pi'$ can therefore be obtained by solving a standard A-optimal design problem depending only on the known feature vectors:
>
>     $$\min_{\pi'\in\Delta_K} \ tr((\sum_{i\in[K]} \pi'(i) x_ix_i^\top )^{-1}) $$ $$ s.t.\ \lambda_{\min}(\sum_{i\in[K]} \pi'(i) x_ix_i^\top ) > 0.$$
>
>     If one further allows $q_t$ to vary, two additional complications arise. First, the safety constraint is no longer automatically satisfied, requiring an explicit cost constraint that cannot be computed directly due to the unknown $c_a$. Second, $\underline{q}$ appears simultaneously in both the estimation error bound and the regret bound, so varying $q_t$ necessitates a joint co-optimization across multiple objectives, which is significantly more complex. We view this problem as a valuable direction for future work and will add a remark in the revised version.
>
> Minor:
> 1. We thank the reviewer for the careful reading. The notation is correct as written. The KL divergence is taken between two probability measures that differ only in the environment $\nu$, with the policy $\pi$ fixed.
>
> 2. It's Average Treatment Effect, defined at its first appearance in Line 015.
>
> Q:
>
> 1. The reviewer's observation is mathematically astute: our policy class $\Pi(\beta, \delta)$ is defined by two constraints, and Thm 3.1 characterizes the minimax estimation error subject to both, so the formulation is indeed equivalent to minimizing estimation error subject to two constraints.
>
>     However, treating regret and safety symmetrically in the lower bound derivation is primarily a matter of analytical convenience for presenting a clean Pareto frontier result. It does not imply they are conceptually equivalent. On the contrary, safety is an operational constraint that must be satisfied regardless — in a clinical trial, no gain in estimation precision justifies compromising patient safety — whereas regret and estimation error are dual performance objectives whose inherent tension constitutes the central focus of this work. This asymmetry motivates our design: safety is enforced as a constraint in both phases, while regret and inference are jointly navigated along the Pareto frontier.
>
> [1] Simchi-Levi et al. Non-stationary Experimental Design under Structured Trends. 2023

---

> > ### Author Rebuttal · Reviewer_yy9W · 2026-04-01
> >
> > Thank you for clarifying my questions. My overall assessment remains the same. The paper makes a valuable contribution in a new setting, although with a limited scope / impact given the motivation remains large theoretical and there are no direct algorithmic implications. The lower bound itself is interesting but not too surprising, and again relies on relatively standard techniques.

---

> > > ### Author Response · Authors · 2026-04-06
> > >
> > > We sincerely thank the reviewer for the time and effort dedicated to evaluating our work, and for the constructive feedback provided throughout the discussion. We appreciate the reviewer's recognition of our contribution. We will carefully incorporate the reviewer's suggestions into the revised manuscript to improve the clarity and presentation of the paper.

---

### Decision · Program_Chairs · 2026-04-30

**Decision:**

Accept (regular)

**Comment:**

This paper studies linear bandit with two simultaneous objectives that need to be traded-off against each other: parameter estimation and regret minimization. Additionally, they assume the presence of safety constraints.

The math seems sound, but the presentation needs some improvements in the exposition of theorem statements. The reviewers are not fully sold on the motivation of the problem. The algorithm and upper bound lacks technical novelty,

Overall, this paper could fit into the conference if there is sufficient space, but it's on the bottom of the priority list.